# SOX2 O-GlcNAcylation alters its protein-protein interactions and genomic occupancy to modulate gene expression in pluripotent cells

Samuel A Myers[1,2†], Sailaja Peddada[3†], Nilanjana Chatterjee[3], Tara Friedrich[4], Kiichrio Tomoda[4], Gregor Krings[5], Sean Thomas[4], Jason Maynard[1], Michael Broeker[6], Matthew Thomson[6], Katherine Pollard[4,7], Shinya Yamanaka[4,8], Alma L Burlingame[1]*, Barbara Panning[3]*

[1]Department of Pharmaceutical Chemistry, University of California, San Francisco, San Francisco, United States; [2]Chemistry and Chemical Biology Graduate Program, University of California, San Francisco, San Francisco, United States; [3]Department of Biochemistry and Biophysics, University of California, San Francisco, San Francisco, United States; [4]Gladstone Institute University of California, San Francisco, San Francisco, United States; [5]Department of Pathology, University of California, San Francisco, San Francisco, United States; [6]Center for Systems and Synthetic Biology, University of California, San Francisco, San Francisco, United States; [7]Institute for Human Genetics, Department of Epidemiology and Biostatistics, University of California, San Francisco, San Francisco, United States; [8]Department of Life Science Frontiers, Center for iPS Cell Research and Application, Kyoto University, Kyoto, Japan

**\*For correspondence:** ALB@cgl. ucsf.edu (ALB); barbara.panning@ gmail.com (BP)

[†]These authors contributed equally to this work

**Abstract** The transcription factor SOX2 is central in establishing and maintaining pluripotency. The processes that modulate SOX2 activity to promote pluripotency are not well understood. Here, we show SOX2 is O-GlcNAc modified in its transactivation domain during reprogramming and in mouse embryonic stem cells (mESCs). Upon induction of differentiation SOX2 O-GlcNAcylation at serine 248 is decreased. Replacing wild type with an O-GlcNAc-deficient SOX2 (S248A) increases reprogramming efficiency. ESCs with O-GlcNAc-deficient SOX2 exhibit alterations in gene expression. This change correlates with altered protein-protein interactions and genomic occupancy of the O-GlcNAc-deficient SOX2 compared to wild type. In addition, SOX2 O-GlcNAcylation impairs the SOX2-PARP1 interaction, which has been shown to regulate ESC self-renewal. These findings show that SOX2 activity is modulated by O-GlcNAc, and provide a novel regulatory mechanism for this crucial pluripotency transcription factor.

## Introduction

SOX2 (sex determining region Y-box 2) is a transcription factor necessary for embryonic stem cell (ESC) self-renewal (*Arnold et al., 2011*; *Masui et al., 2007*). Precise control of SOX2 is critical for ESC maintenance, since increased or decreased expression of SOX2 interferes with self-renewal and pluripotency (*Kopp et al., 2008*; *Masui et al., 2007*). Post-translational modifications (PTMs) of SOX2 may play a role in its regulation, as SOX2 is reported to be phosphorylated, methylated, ubiquitinylated, SUMOylated, acetylated, and PARylated (*Baltus et al., 2009*; *Brumbaugh et al., 2012*;

**eLife digest** Embryos develop from stem cells, which have the ability to mature into any type of cell in the body. The activity of proteins called transcription factors determines whether a stem cell will become a specialized cell type or remain in an immature "pluripotent" state that has the potential to become any cell type. These transcription factors bind to the cell's DNA to regulate the activity of target genes.

SOX2 is a transcription factor that helps to maintain embryonic stem cells in a pluripotent state. In 2011, a group of researchers showed that a specific sugar molecule was added to SOX2 in mouse embryonic stem cells, in a process called *O*-GlcNAcylation. Now, Myers, Peddada et al. – including the researchers who performed the 2011 study – have studied the effects of this SOX2 modification in more detail.

Transcription factors have two major activities – they bind to DNA and recruit other proteins that can turn target genes on or off. Myers, Peddada et al. found that, in pluripotent stem cells, a complex pattern of *O*-GlcNAcylation is present on SOX2 in a region that is responsible for recruiting other proteins. In addition, SOX2 *O*-GlcNAcylation decreases when stem cells are directed to become a new cell type.

Further experiments investigated gene activity in stem cells that contained a mutant form of SOX2 that cannot be *O*-GlcNAc modified. In these cells, genes that help to maintain the cell in a pluripotent state were more active than in normal cells. The mutant form of SOX2 was altered in its ability to bind DNA and to associate with proteins that control gene activity.

Myers, Peddada et al.'s findings raise several questions. Does *O*-GlcNAcylation control the activity of SOX2 in other cell types, such as neurons and cancer cells, in which this modification can be detected on SOX2? Why does a modification on the portion of the SOX2 that is thought to interact with other proteins affect SOX2 DNA binding activity? Finally, understanding how *O*-GlcNAcylation is employed to regulate SOX2 activity in response to developmental cues remains a major challenge.

*Fang et al., 2014*; *Gao et al., 2009*; *Lai et al., 2012*; *Swaney et al., 2009*; *Tahmasebi et al., 2013*; *Tsuruzoe et al., 2006*; *Van Hoof et al., 2009*; *Zhao et al., 2011*).

We have previously shown SOX2 is *O*-linked *N*-acetlyglucosamine (*O*-GlcNAc) modified in mouse ESCs (mESCs) (*Myers et al., 2011*). *O*-GlcNAcylation is the dynamic and regulatory mono-glycosylation of nucleocytosolic proteins catalyzed by a single *O*-GlcNAc transferase (OGT) and removed by a single hydrolase (OGA/MGEA5/NCOAT). *O*-GlcNAc signaling is essential for embryo viability (*O'Donnell et al., 2004*; *Shafi et al., 2000*; *Yang et al., 2012*) and mESC self-renewal (*Jang et al., 2012*). While OGT is critical for mESC maintenance, the protein- and site-specific functions of *O*-GlcNAcylation in mESCs have not been fully elucidated.

Here, we show that *O*-GlcNAcylation of SOX2 at serine 248 (S248) is dynamically regulated in mESCs. Upon differentiation, *O*-GlcNAc occupancy is reduced and SOX2 is predominantly unmodified at this site. Replacement of wild type SOX2 (SOX2$^{WT}$) with an *O*-GlcNAc-deficient mutant SOX2 (SOX2$^{S248A}$) results in increased reprogramming efficiency. mESCs with SOX2$^{S248A}$ as their sole source of SOX2 have increased expression of genes associated with pluripotency and exhibit a decreased requirement for OCT4. SOX2$^{S248A}$ exhibits altered genomic occupancy and differential association with transcriptional regulatory complexes. *O*-GlcNAcylation directly inhibits the SOX2 and PARP1 interaction, which plays a regulatory role in mESC pluripotency (*Gao et al., 2009*; *Lai et al., 2012*). This study implicates *O*-GlcNAc modification in coordinating genomic occupancy and protein-protein interactions of SOX2 in ESCs, and provides molecular insight into how this broadly expressed transcription factor is regulated to promote the pluripotency-specific expression program.

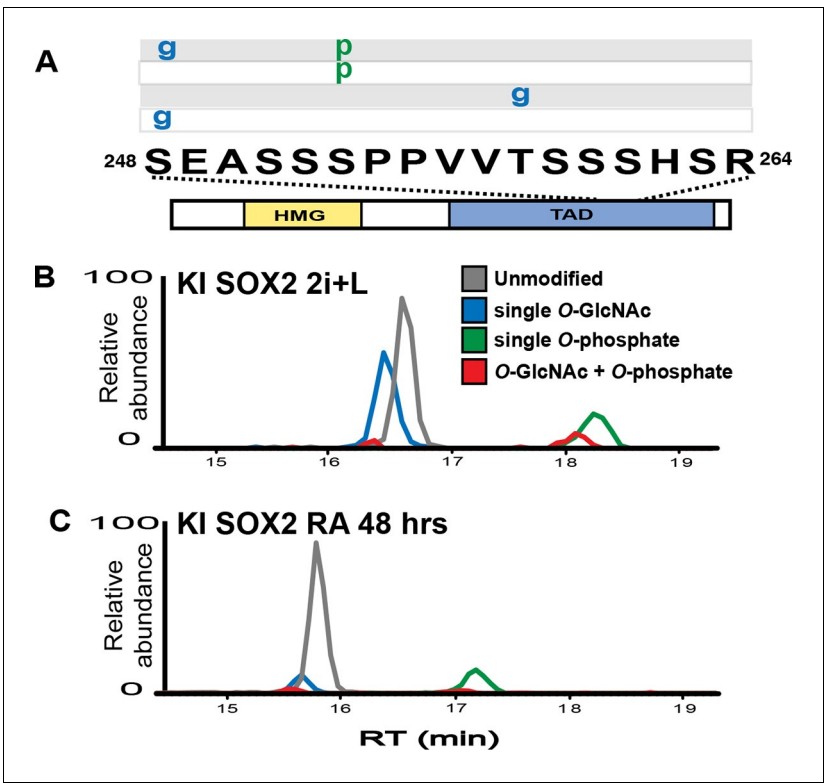

**Figure 1.** SOX2 *O*-GlcNAc levels change during differentiation. (**A**) Diagram of SOX2 (bottom, with TAD and high mobility group DNA binding domain [HMG] indicated), the TAD peptide sequence (middle; amino acid numbering from the Uniprot accession number P48432), and the PTM isoforms identified on the TAD peptide (top, grey and white rectangles, g indicates *O*-GlcNAc and p indicates phosphate). Mass spectra can be seen in *Figure 1—figure supplement 1*. (**B**) and (**C**) Extracted ion chromatographs (XICs) of SOX2 TAD peptide PTM states from (**B**) undifferentiated KI SOX2 mESCs (2i+L) or (**C**) differentiated KI SOX2 mESCs (RA 48 hr). Traces for each PTM isoform are colored differently, key provided in the inset in (**B**).

The following figure supplements are available for figure 1:

**Figure supplement 1.** ETD-MS/MS spectra of SOX2 unmodified TAD peptide described in *Figure 1A*.

**Figure supplement 2.** ETD-MS/MS spectra of SOX2 GlcNAc-S248 TAD peptide described in *Figure 1A*.

**Figure supplement 3.** ETD-MS/MS spectra of SOX2 GlcNAc-T258 TAD peptide described in *Figure 1A*.

**Figure supplement 4.** ETD-MS/MS spectra of SOX2 phospho-S253 TAD peptide described in *Figure 1A*.

**Figure supplement 5.** ETD-MS/MS spectra of SOX2 co-modified GlcNAc-S248/phospho-S253 TAD peptide described in *Figure 1A*.

**Figure supplement 6.** *O*-GlcNAcylation of OCT4 at T228 is undetectable in mESCs.

**Figure supplement 7.** Incorrect assignment of GluC digest OCT4 peptide mass spectrum containing T228.

## Results

### SOX2 *O*-GlcNAcylation is regulated by differentiation cues

Previously, we reported in mESCs SOX2 was *O*-GlcNAcylated in the transactivation domain (TAD) (*Myers et al., 2011*), a region described to possess several other PTMs (*Brumbaugh et al., 2012*; *Swaney et al., 2009*; *Tahmasebi et al., 2013*; *Tsuruzoe et al., 2006*; *Van Hoof et al., 2009*). To

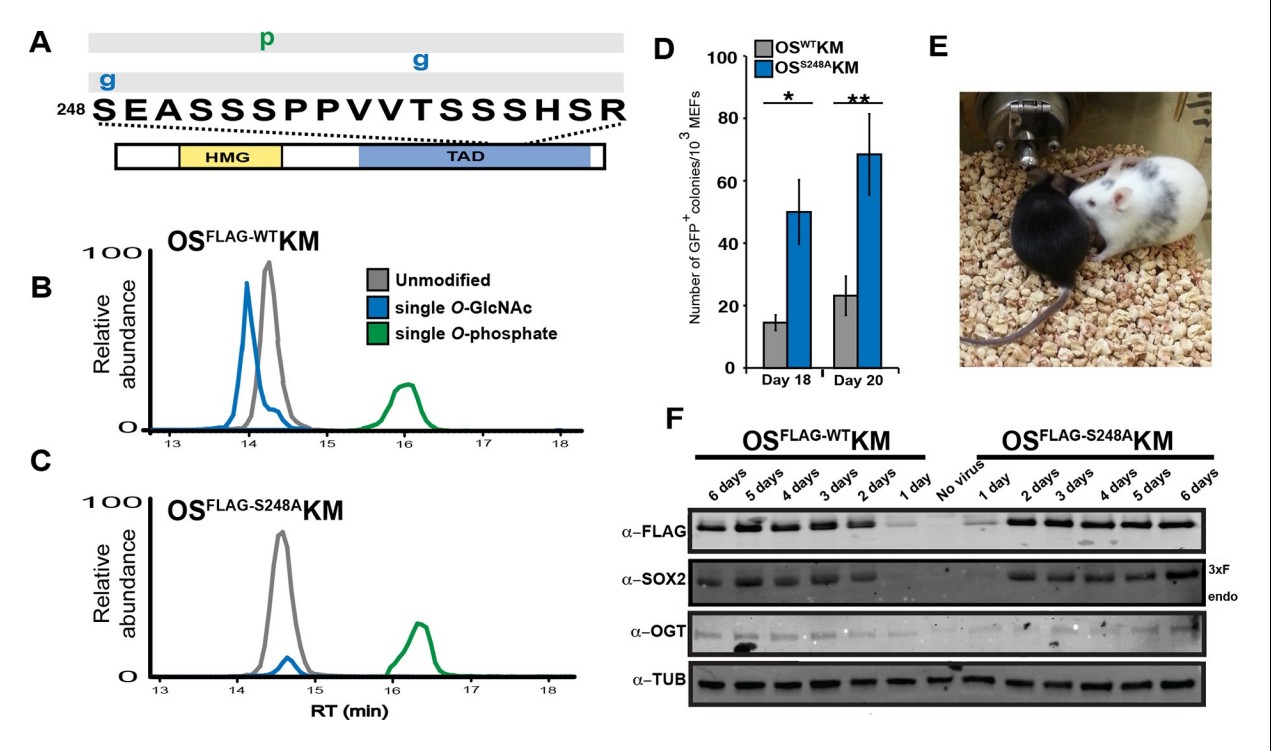

**Figure 2.** *O*-GlcNAc-deficient SOX2, SOX2[S248A], increases somatic cell reprogramming efficiency. (**A**) Diagram of SOX2 and the PTMs identified from MEFs transduced with OS[FLAG-WT]KM, labeled as described in *Figure 1A*. Spectra can be found at *tinyurl.com/iPSC-3xF-SOX2-ETD* and *tinyurl.com/ iPSC-3xF-SOX2-HCD*. (**B**) XICs of 3xF-SOX2[WT] TAD peptide PTM states from OS[FLAG-WT]KM-transduced MEFs. (**C**) XICs of 3xF-SOX2[S248A] TAD peptide PTM states from OS[FLAG-S248A]KM-transduced MEFs. Color key the same as in (**B**). (**D**) Number of GFP[+] colonies from 1000 *Nanog-Gfp* MEFs transduced with OS[WT]KM or OS[S248A]KM and cultured on SNL feeders for 18 or 20 days (n=7 +/- S.E.M.). (**E**) Chimeric mouse derived from iPSCs obtained from transducing *Nanog-Gfp* MEFs with OS[S248A]KM and his black offspring, demonstrating germline transmission. (**F**) Western blots against FLAG, SOX2, OGT and TUBULIN for the first six days of reprogramming with either OS[FLAG-WT]KM or OS[FLAG-S248A]KM. "Endo" refers to the apparent molecular weight at which the endogenous SOX2 would be expected, "3xF" refers the the FLAG tagged version from the viral transduction.

The following figure supplements are available for figure 2:

**Figure supplement 1.** Immunofluorescence staining against FLAG in MEFs six days after transduction with either OS[FLAG-WT]KM or OS[FLAG-S248A]KM shows similar nucleocytoplasmic distribution.

**Figure supplement 2.** SOX2[S248D] also increases somatic cell reprogramming efficiency.

investigate whether PTMs in the TAD are subject to developmental regulation, we analyzed SOX2 during the initial stages of differentiation. Knock-in FLAG/HA tagged SOX2 mESCs (KI cells; *Lai et al., 2012*) were cultured in media containing MEK and GSK3b inhibitors (2i) and leukemia inhibitory factor (LIF), or were induced to differentiate by removing LIF and adding retinoic acid (RA). Liquid chromatography coupled with tandem mass spectrometric (LC-MS/MS) analysis of SOX2 in self-renewal conditions revealed four main populations of the TAD peptide containing serine 248 (hereinafter referred to as TAD peptide): unmodified, *O*-GlcNAcylated at one of two sites (S248 or T258), phosphorylated at S253, and doubly modified with *O*-GlcNAcylation at S248 and phosphorylation at S253, (*Figure 1A—figure supplements 1–5*). Removal of LIF and addition of RA for 48 hr resulted in a marked decrease in the *O*-GlcNAc occupancy of the TAD peptide with no change in the phosphorylation stoichiometry (*Figure 1B–C*). These data indicate *O*-GlcNAcylation of SOX2 S248 is responsive to differentiation cues.

In mESCs SOX2 heterodimerizes with OCT4, which is also reported to be *O*-GlcNAcylated in this cell type (*Jang et al., 2012*). Thus, it is possible that OGT targets both these transcription factors

when they are complexed together, prompting us to query OCT4 O-GlcNAcylation. However, we were unable to detect OCT4 O-GlcNAcylation in mESCs (*Figure 1—figure supplements 6–7*).

## O-GlcNAc-deficient SOX2 increases somatic cell reprogramming efficiency

To query whether O-GlcNAcylation at S248 is present in other contexts, we examined the PTM profile of the SOX2 TAD peptide during somatic cell reprogramming. We used four-factor retroviral reprogramming (*Oct4, Sox2, Klf4* and *Myc*; OSKM) in which SOX2 contained a triple FLAG tag (OS$^{FLAG-WT}$KM). LC-MS/MS analysis of purified SOX2 six days after transduction of mouse embryonic fibroblasts (MEFs) showed S248 is O-GlcNAcylated (*Figure 2A*). In addition, mutation of S248 to alanine (S248A) resulted in loss of O-GlcNAcylation without affecting the other PTMs of the TAD peptide (*Figure 2B–C*). These results demonstrate S248 is O-GlcNAc modified during somatic cell reprogramming and suggest a connection between this SOX2 PTM and pluripotency.

To determine whether the S248A mutation impacted induced pluripotent stem cell (iPSC) colony formation, we used somatic cell reprogramming of *Nanog-Gfp* reporter MEFs (*Takahashi and Yamanaka, 2006*). *Nanog-Gfp* MEFs transduced with OS$^{S248A}$KM produced significantly more GFP$^+$ iPSC colonies compared to OS$^{WT}$KM (*Figure 2D*). iPSCs generated with OS$^{S248A}$KM exhibited standard colony morphology and contributed to chimeric mice capable of germ line transmission (*Figure 2E*), indicating these OS$^{S248A}$KM iPSCs exhibit the features of normal iPSCs. By Western blot and immunostaining of MEFs transduced with OS$^{FLAG-WT}$KM or OS$^{FLAG-S248A}$KM showed equal levels of

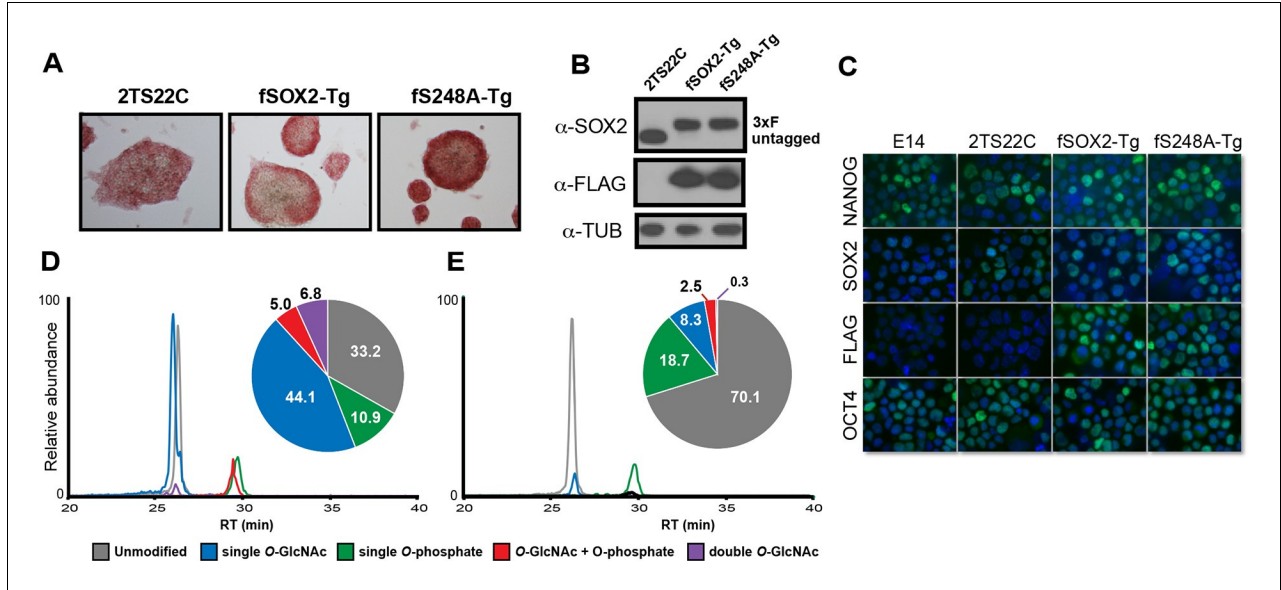

**Figure 3.** SOX2$^{S248A}$ can replace wild type SOX2 in mESCs. (**A**) Characterization of fSOX2-Tg and fS248A-Tg mESCs. fSOX2-Tg and fS248A-Tg mESCs exhibit AP staining, a marker of pluripotency, similar to parental 2TS22C cells. (**B**) Western blot analysis of SOX2 and FLAG in 2TS22C, fSOX2-Tg and fS248A-Tg mESCs. TUBULIN (TUB) is used as a loading control. "3xFLAG" and "untagged" refer to expected molecular weights of SOX2 with the 3xFLAG tag or no tag, respectively. (**C**) Immunofluorescence staining for NANOG, SOX2, FLAG and OCT4 in wild type E14, parental 2TS22C, fSOX2-Tg, and fS248A-Tg mESCs. Antibody staining is green, nuclear stain with DAPI is blue. (**D**) and (**E**) XICs of the TAD peptides of SOX2 immunopurified from fSOX2-Tg (**D**) and fS248A-Tg (**E**) mESCs. Insets: pie charts showing the mean percentage of each PTM form to total TAD peptide signal (n=3). The doubly phosphorylated TAD peptide is below the limit of quantitation for both cell lines.

The following figure supplements are available for figure 3:

**Figure supplement 1.** Diagram of creation of fSOX2-Tg or fS248A-Tg lines.

**Figure supplement 2.** Diagram of SOX2 and the PTMs identified from fSOX2-Tg cells, labeled as described in *Figure 1A*.

**Figure supplement 3.** Label-free MS1 analysis of synthetic SOX2 TAD peptides.

exogenous SOX2 for the first six days ofof reprogramming (*Figure 2F* and *Figure 2—figure supplement 1*), indicating comparable expression of WT and S248A triple FLAG tagged SOX2. OGT levels were also similar for the first six days of reprogramming between OS$^{FLAG-WT}$KM and OS$^{FLAG-S248A}$KM transduced MEFs (*Figure 2F*). These results indicate that SOX2$^{S248A}$ is more efficient than wild type SOX2 at inducing pluripotency and suggest *O*-GlcNAcylation at S248 inhibits SOX2 activity.

The homologous SOX2 residue has been reported to be phosphorylated in human ESCs (*Swaney et al., 2009*). While our lab and others were unable to detect this phosphorylation in mESCs (*Brumbaugh et al., 2012*) or during murine reprogramming, the S248A mutation could potentially remove a regulatory phosphorylation site. Therefore, we performed reprogramming experiments using the phospho-mimetic SOX2 mutant, S248D. This mutation also increased reprogramming efficiency (*Figure 2—figure supplement 2*), suggesting it is the removal of an *O*-GlcNAcylation site, and not of a phosphorylation site, which mediates the effect on SOX2 activity.

## SOX2$^{S248A}$ can replace wild type SOX2 in mESCs

Since the reprogramming results suggested the S248A mutation increased SOX2 activity, we examined whether this mutant SOX2 supported mESC self-renewal. We generated mESC lines that express either a FLAG-tagged wild-type *Sox2* transgene (fSOX2-Tg cells) or an S248A transgene (fS248A-Tg cells) (*Figure 3A*). We introduced the transgenes into 2TS22C mESCs, in which endogenous *Sox2* is removed and a doxycycline repressible SOX2 cDNA transgene supports self-renewal (*Masui et al., 2007*)(*Figure 3—figure supplement 1*). Under doxycycline repression, the sole source of SOX2 in these transgenic lines is the FLAG-tagged wild-type or S248A mutant SOX2 (*Figure 3B*). SOX2 levels in fSOX2-Tg and fS248A-Tg mESCs are comparable to SOX2 levels in the 2TS22C parental cell line and nucleo-cytoplasmic distribution was not altered by the mutation (*Figure 3C*). OCT4 and NANOG abundance and distribution were comparable between fSOX2-Tg and fS248A-Tg mESCs (*Figure 3C*), arguing that there is no gross effect on these pluripotency transcription factors.

LC-MS/MS analysis of immunopurified SOX2 from fSOX2-Tg mESCs identified nine PTM forms of the SOX2 TAD peptide (*Figure 3—figure supplement 2*). LC-MS analysis of the TAD peptide precursor masses from fSOX2-Tg mESCs showed unmodified and singly *O*-GlcNAcylated were the most abundant forms of the SOX2 TAD peptide (33.2 and 44.1% of total TAD, respectively) (*Figure 3D*). LC-MS analysis confirmed the loss of S248 *O*-GlcNAcylation in fS248A-Tg mESCs (*Figure 3E*). Analysis of synthetic SOX2 TAD peptides showed chromatographic separation of PTM or mutant isoforms, and lack of electrospray ionization suppression, validating our label free quantitation approach (*Figure 3—figure supplement 3*). In addition, the TAD peptide in fS248A-Tg mESCs showed increased phosphorylation at S253, from 10.9 to 18.7% of total TAD, suggesting cross talk between phosphorylation and *O*-GlcNAcylation.

## SOX2$^{S248A}$ alters gene expression in mESCs

To determine if the S248A mutation altered global transcript levels we used microarrays to compare the gene expression profiles of fSOX2-Tg and fS248A-Tg mESCs. Significant changes in mRNA levels were observed, with 320 genes up regulated and 344 genes down regulated in fS248A-Tg cells (*Figure 4A*) and gene set enrichment analysis of differentially expressed genes did not show significant enrichment of any pathways. Several genes that promote pluripotency and self-renewal were upregulated, while several genes associated with differentiation were down-regulated in fS248A-Tg cells compared to WT. RT-qPCR confirmed the differential expression of these pluripotency or differentiation genes (*Figure 4B*). These data suggest the S248A mutation alters the balance between self-renewal and differentiation gene expression in mESCs.

The altered gene expression profile of fS248A-Tg cells suggested this mutation may promote self-renewal at the expense of differentiation. Therefore, we examined the effects of OCT4 depletion, which causes mESCs to differentiate (*Figure 4C*) (*Hough et al., 2006*). While fSOX2-Tg mESCs exhibited altered cell and colony morphology (*Figure 4—figure supplement 1*), decreased AP staining (*Figure 4D*), and decreased expression of *Nanog* (*Figure 4E*), fS248A-Tg mESCs were relatively unaffected by OCT4 depletion. These data indicate that fS248A-Tg mESCs can maintain key features

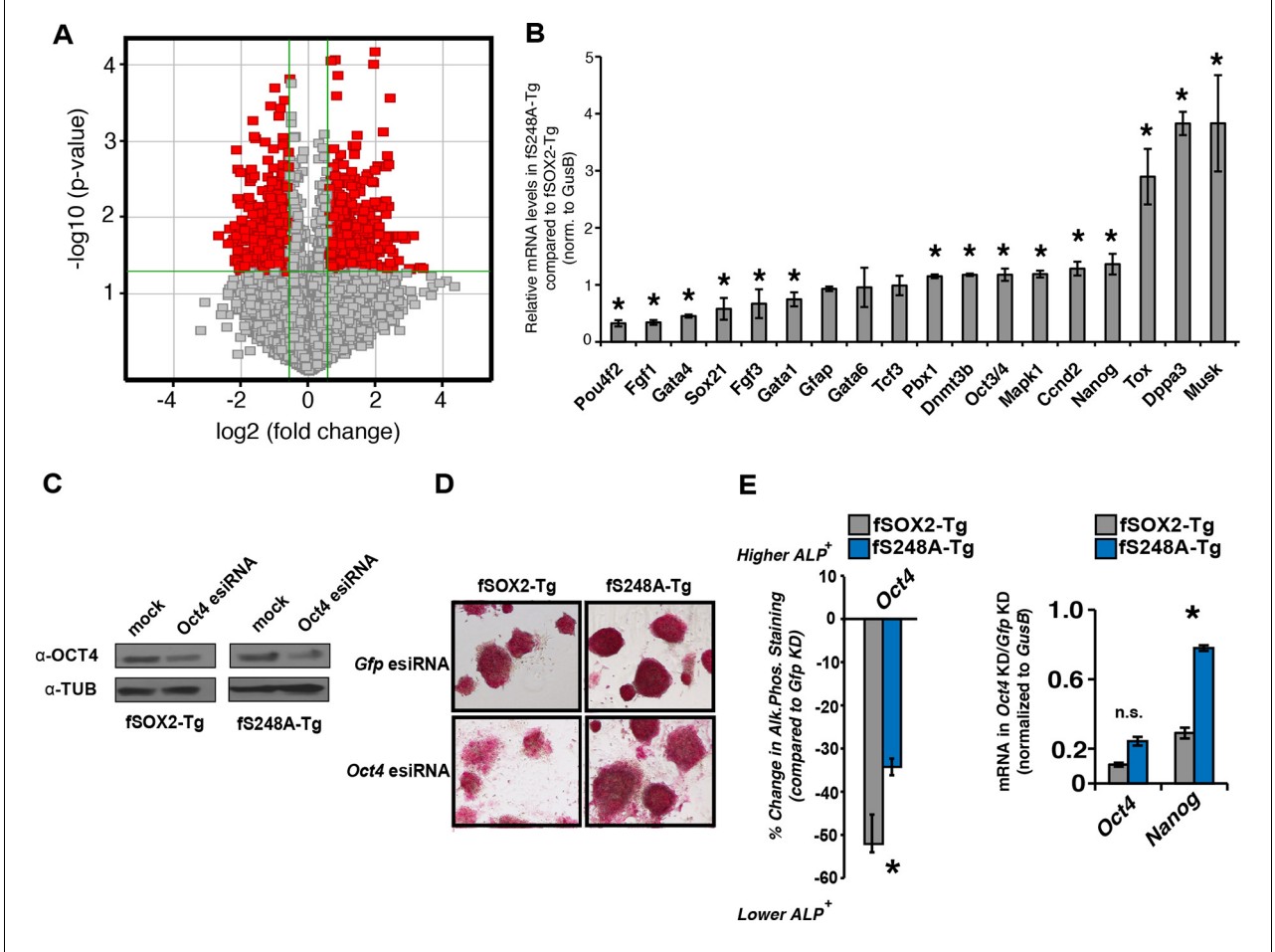

Figure 4. fS248A-Tg mESCs show altered gene expression and decreased dependence on OCT4. (A) Volcano plot of global changes in gene expression between fSOX2-Tg and fS248A-Tg cells. Red indicates genes with increased or decreased expression (fold change cutoff 1.5 and paired t-test p<0.05) (**Supplementary file 1a**). (B) RT-qPCR of select genes differentially expressed between fSOX2-Tg and fS248A-Tg cells (* indicates p<0.05, n=3, +/- S.E.M.). (C) fSOX2-Tg or fS248A-Tg cells were depleted of OCT4 using siRNA pools (esiRNAs) and Western blot analysis of OCT4 and TUBULIN were performed. (D) and (E), (D) AP staining and (E) quantitation of fold change in AP staining three days after OCT4 or GFP depletion in fSOX2-Tg and fS248A-Tg cells. Additional example fields of view for relative quantitation can be seen in **Figure 4—figure supplement 1**. F, RT-qPCR analysis of *Oct4* and *Nanog* mRNA levels in fSOX2-Tg or fS248A-Tg cells depleted of OCT4 compared to the control knockdown of GFP.

The following figure supplement is available for figure 4:

**Figure supplement 1.** AP activity staining of fSOX2-Tg and fS248A-Tg cells three days after *Gfp* or *Oct4* knockdown.

of pluripotency when OCT4 levels are reduced, and are consistent with a role for the *O*-GlcNAc modification inhibiting SOX2 activity.

## SOX2$^{S248A}$ exhibits altered genomic occupancy

To examine whether the altered gene expression associated with the S248A mutation was accompanied by changes in SOX2 genomic occupancy, we performed FLAG chromatin immunoprecipitation followed by next generation sequencing (ChIP-seq) to compare SOX2 genomic distribution in fSOX2-Tg and fS248A-Tg mESCs (**Figure 5A**). SOX2 distribution exhibited considerable overlap, with 4,191 sites bound in both lines (**Figure 5B**). The mutant form of SOX2 occupied 1000 sites not bound by the wild type form (**Figure 5A**). *De novo* motif analysis identified the SOX2 binding motif in fS248A-Tg specific peaks (**Figure 5C**). In mESCs, SOX2 and OCT4 heterodimerize and co-occupy a substantial portion of their target regulatory sequences (**Boyer et al., 2005**). *De novo* motif

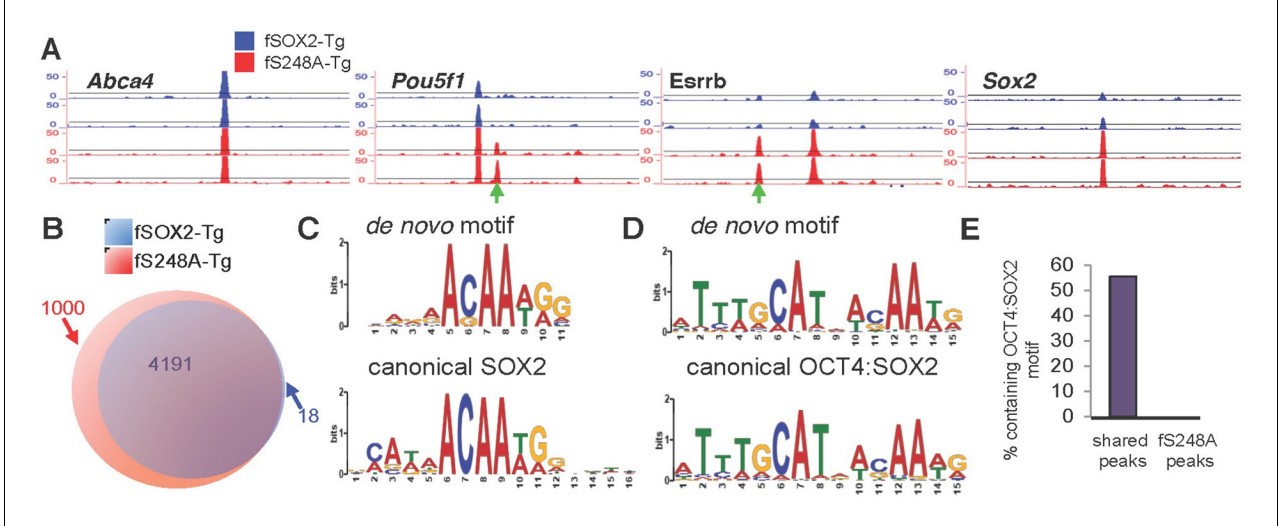

**Figure 5.** S248A mutation alters genome-wide distribution of SOX2. (**A**) Representative UCSC genome browser tracks of FLAG ChIP-seq in fSOX2-Tg (blue) and fS248A-Tg (red) cells. Examples of fS248A-Tg specific peaks (*Pou5f1, Esrrb*) and shared peaks (*Abca4, Sox2*) are shown for 2 biological replicates (2 technical replicates were performed for each biological replicate, Spearman correlations for technical replicates are 1, for biological replicates 0.45 for fSOX2-Tg and 0.55 for fS248A-Tg). Each track is 15 kb. Green arrows indicate fS248A-Tg specific peaks. For *Sox2* track, the region shown is not encompassed in the deletion removing endogenous *Sox2*. (**B**) Overlap (purple) in called peaks from anti-FLAG ChIP-seq in fSOX2-Tg (blue) and fS248A-Tg (red) mESCs. (**C**) *De novo* SOX2 motif identified in shared ChIP-seq peaks between fSOX2-Tg and fS248A-Tg cells (top) compared to the canonical SOX2 motif [Jaspar M01271] (bottom). (**D**) OCT4:SOX2 motif identified in peaks shared between fSOX2-Tg and fS248A-Tg cells using *de novo* motif analysis (top) compared to the canonical OCT4:SOX2 motif [Jaspar MA0142.1] (bottom). (**E**) Proportion of peaks containing a motif matching the OCT4:SOX2 *de novo* motif in shared peaks (left) and fS248A-Tg specific peaks (right).

analysis of SOX2 peaks shared between fSOX2-Tg and fS248A-Tg mESCs identified the OCT4:SOX2 motif (*Figure 5D*), which was present in 2335 of the shared peaks. The OCT4:SOX2 motif was not identified in any of the fS248A-Tg-specific peaks (*Figure 5E*). These data indicate the S248A mutation alters SOX2 genomic distribution, increasing its ability to associate with SOX2 binding sites that would not ordinarily be bound by wild type SOX2 in mESCs.

## *O*-GlcNAc alters SOX2 protein-protein interactions

S248 lies in the TAD of SOX2, a region responsible for interactions with transcriptional regulatory machinery (*Ambrosetti et al., 2000*; *Nowling et al., 2000*; *Yuan et al., 1995*). Therefore, we tested whether the S248A mutation altered SOX2 centered protein-protein interactions (PPIs). We performed affinity purifications against FLAG from nuclear extracts of fSOX2-Tg, fS248A-Tg or an equivalent mESC line expressing transgenic HA-tagged SOX2 (haSOX2-Tg, *Figure 6—figure supplement 1*) and used quantitative LC-MS to identify proteins that co-purified with FLAG in each cell type. We identified 329 proteins enriched in both fSOX2-Tg and fS248A-Tg, but not haSOX2-Tg FLAG IPs. Many of these interactors exist in complexes involved in histone modification, DNA damage repair, or nucleosome remodeling (*Figure 6A*). Several SOX2 interactors have been previously described (*Cox et al., 2013*; *Engelen et al., 2011*; *Gao et al., 2012*), indicating fSOX2-Tg and fS248A-Tg cells recapitulate some known SOX2 interactions (*Supplementary file 1b*).

We next examined whether any co-purifying proteins were enriched in either the fSOX2-Tg or fS248A-Tg co-IPs, by plotting the enrichment ratios between S248A and wild type. 22 of the interacting proteins were enriched at least four-fold in the co-IP with fSOX2-Tg (z-score > 1.5) and 60 were enriched in the fS248A-Tg co-IP (*Supplementary file 1b*). Co-IP followed by Western blotting corroborated the IP-MS data, showing preferential enrichment of PARP1 and GATAD2B with mutant and wild type SOX2, respectively, while SMARCA4 was associated equally with both forms of SOX2 (*Figure 6B*).

Examination of the protein complexes enriched by either wild type or S248A SOX2 showed a subset of components behaved discordantly with the rest of the complex subunits. For example, MBD3

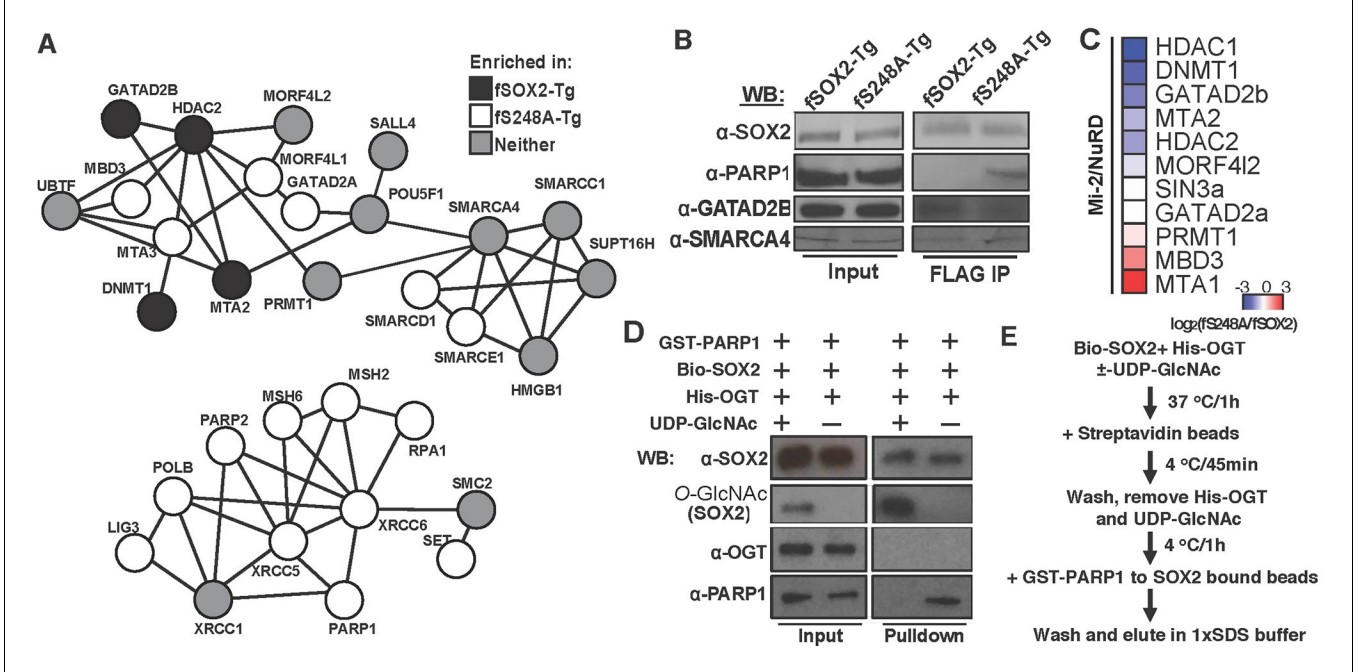

**Figure 6.** *O*-GlcNAcylation of SOX2 at S248 alters protein-protein interactions. (A) Interaction diagram of a subset of SOX2 interactors that exhibit differential association with 3xF-SOX2^S248A relative to 3xF- SOX2^WT. Color of circles indicates with which SOX2 proteoform a protein preferentially interacts. Interaction diagram based on high confidence, experimental interactions identified by STRING. (B), Anti-FLAG IP-WB for SOX2, PARP1, GATAD2B, and SMARCA4 in fSOX2-Tg and fS248A-Tg cells. (C) Heatmap of median enrichment values of NuRD subunits that preferentially associate with 3xF-SOX2^WT or 3xF-SOX2^S248A as determined by quantitative mass spectrometry (n=3). (D) Western blot analysis of in vitro interaction between SOX2 +/- *O*-GlcNAcylation and PARP1. Bio-SOX2 and His-OGT were incubated with and without UDP-GlcNAc, Bio-SOX2 purified away from OGT and UDP-GlcNAc using streptavidin beads and incubated with GST-PARP1. Western blots examine proteins associated with streptavidin beads. Comparable amounts of input and pull down were loaded for all blots, except *O*-GlcNAc, in which more material was loaded in the pull down lanes. WB, Western blot; GST, glutathione S-transferase tag; Bio, biotinylated Bio tag; His, polyhistidine tag. (E) Flow chart outlining scheme for **D**.

The following figure supplements are available for figure 6:

**Figure supplement 1.** Creation of haSOX2-Tg, where HA-tagged SOX2 is the sole source of SOX2.

**Figure supplement 2.** *O*-GlcNAc site mapping by ETD-MS/MS of recombinant Bio-tagged human SOX2 incubated with recombinant human OGT and UDP-GlcNAc.

and MTA3, both of which can be a part of the NuRD complex, were consistently enriched in fS248A-Tg co-IPs while other NuRD components were enriched with fSOX2-Tg (*Figure 6B*). To more thoroughly investigate the subunit distribution of a subset of the NuRD complex, we used a targeted proteomic approach based on interacting proteins from an MBD3 co-IP experiment. We performed anti-FLAG affinity purifications in FLAG tagged MBD3 mESCs (*Yildirim et al., 2011*) followed by LC-MS to generate a representative set of NuRD complex peptides. The top two, best scoring, unique peptides for each NuRD component were used to determine the relative enrichment of these proteins from fSOX2-Tg and fS248A-Tg co-IPs (*Supplementary file 1c*). Targeted analysis showed the majority of the NuRD complex preferentially associated with SOX2^WT, while MBD3 and MTA3 components prefer SOX2^S248A (*Figure 6C*). These results suggest that the S248A mutation can affect the stoichiometry of subunits in complexes that associate with SOX2.

The altered PPIs with SOX2^S248A may occur as a direct result of the lack of *O*-GlcNAcylation of S248. To test whether the *O*-GlcNAcylation of SOX2 was directly responsible for alterations in a PPI, we used recombinant proteins to assess the effect of this PTM on the SOX2-PARP1 interaction. *O*-GlcNAcylation of Bio-tagged, recombinant human SOX2 (Bio-SOX2, 96% identical to mouse) by recombinant human OGT (His-OGT, 99% identical to mouse), which depended on the sugar donor UDP-GlcNAc, was detected by Western blotting (*Figure 6D*) and specificity confirmed by mass spec

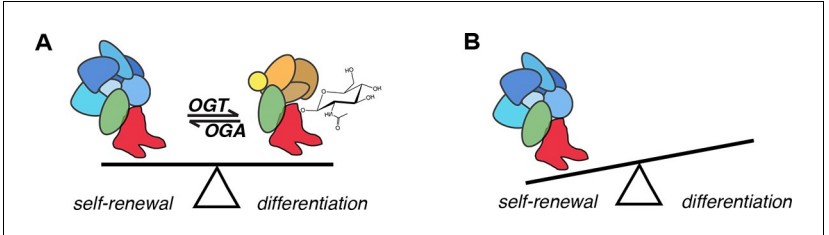

**Figure 7.** Model for the role of *O*-GlcNAcylation in regulation of SOX2 in mESCs. (**A**) *O*-GlcNAc (sugar moiety) affects the affinity of SOX2 (red) for interacting proteins (ovals). Some proteins (blue shapes) exhibit greater affinity for unmodified SOX2, while others exhibit lower affinity (orange shapes). In addition, *O*-GlcNAcylation affects SOX2 binding to a subset of target DNA sequences. (**B**) As a result of altered genomic distribution and protein-protein interactions when SOX2 cannot be *O*-GlcNAcylated (SOX2^S248A), pluripotency gene expression is promoted at the expense of differentiation.

(*Figure 6—figure supplement 2*). Bio-SOX2 was bound to streptavidin magnetic beads to remove OGT and UDP-GlcNAc. Beads bound by *O*-GlcNAcylated or unmodified SOX2 were incubated with GST-tagged, recombinant human PARP1 (GST-PARP1, 91% identical to mouse) (*Figure 6E*). Pull down efficiency of Bio-SOX2 was not affected by *O*-GlcNAcylation and His-OGT was not detected in pull downs, indicating any potential SOX2:OGT interaction was not stable under our wash conditions. Unmodified Bio-SOX2 pulled down GST-PARP1, indicating the interaction between mouse SOX2 and PARP1 can be recapitulated by their conserved human homologues. Pulldown efficiency of GST-PARP1 by glycosylated Bio-SOX2 was diminished compared to that of unmodified Bio-SOX2 (*Figure 6D*). Together, these data demonstrate SOX2 *O*-GlcNAcylation directly alters its interaction with a transcriptional regulatory protein involved in maintaining the balance of self-renewal and differentiation (*Figure 7*).

## Discussion

Depletion of OGT, the sole enzyme that mediates intracellular *O*-GlcNAcylation, disrupts mESC self-renewal (*O'Donnell et al., 2004*; *Shafi et al., 2000*), prompting us to identify OGT targets to elucidate link between *O*-GlcNAc and self-renewal. Using an unbiased strategy for enrichment of native *O*-GlcNAcylated nuclear peptides, we previously identified SOX2 S248 as an OGT substrate (*Myers et al., 2011*). Here, we find that S248 is *O*-GlcNAcylated during somatic cell reprogramming and that mutation of this residue to alanine increases reprogramming efficiency. We also find mESCs expressing SOX2^S248A exhibit changes in transcription consistent with increased expression of pluripotency promoting genes at the expense of differentiation promoting genes. Together, these analyses from both mESCs and during somatic cell reprogramming reveal the S248A mutation promotes SOX2 activity, which suggests SOX2 *O*-GlcNAcylation is inhibitory during maintenance and establishment of pluripotency.

Our data indicate that S248 *O*-GlcNAcylation is regulated by developmental signaling molecules, since removing LIF and adding RA to trigger differentiation resulted in a substantial decrease in this PTM. This decrease in S248 *O*-GlcNAcylation appears contradictory to the finding that the S248A mutation, which eliminates *O*-GlcNAcylation, promotes mESC self-renewal. However, the effects of this, or any, PTM are likely to be context specific, and determined by the transcription factors and signaling molecules present in each cell type. GlcNAc-S248 may inhibit SOX2 activity in mESCs, where the decrease in this PTM may alter SOX2 activity upon differentiation, such that SOX2 functions appropriately for the changing cellular context. As this work shows, use of methods that allow analysis of SOX2 PTM-specific PPIs and genomic occupancy may be crucial to understand how combinations of PTMs are used to regulate SOX2 activity in response to developmental cues.

In addition to changes in gene expression, the S248A mutation altered SOX2 genomic distribution. As well as occupying the same sites as SOX2^WT, SOX2^S248A was found at an additional 1000 sites. The majority of these sites contained a predicted SOX2 binding motif, indicating the mutation allows SOX2^S248A to occupy sites that SOX2^WT is unable to access in mESCs. This result suggests *O*-GlcNAcylation can regulate the affinity of SOX2 for its target sites. Since the mutation lies in the

TAD, but affects the activity of the high mobility group DNA binding region, the mutation or loss of *O*-GlcNAcylation could affect secondary protein structure and/or PPIs. Single molecule imaging of SOX2 with a deleted TAD showed altered DNA occupancy and higher site-specific residence time, supporting the idea that TAD mutations affect SOX2 DNA binding (*Chen et al., 2014*).

Upon OCT4 knockdown, fS248A-Tg mESCs did not exhibit as dramatic a change in colony morphology as fSOX2-Tg mESCs. In addition, the new sites of genomic occupancy seen in fS248A-Tg mESCs did not have nearby predicted OCT4 motifs. Together these results suggest the *O*-GlcNAc-deficient SOX2 may exhibit an altered reliance on OCT4 for binding and regulation of gene expression. Further studies aimed at querying the effects of the S248A mutation on genome-wide OCT4 distribution should provide insight into whether this SOX2 PTM alters OCT4 association with target sequences, and may address if *O*-GlcNAc impacts OCT4/SOX2 heterodimerization.

Our proteomic analyses indicated there are substantial differences in SOX2-centered PPIs in fS248A-Tg mESCs. Many of the proteins that exhibited differential interaction between wild type and mutant SOX2 are components of complexes implicated in chromatin regulation. These altered associations may underlie the transcriptional and genomic occupancy changes seen in the mutant mESCs. In addition, components of the PARP-XRCC and the DNA mismatch repair (MMR) complexes were enriched with SOX2$^{S248A}$, suggesting the possibility that these complexes may function in transcriptional regulation in addition to DNA repair. Consistent with this hypothesis, DNA damage complexes promote mESC self-renewal and iPSC generation (*Fong et al., 2011*).

Our PARP1 results are consistent with previous reports describing its interaction with SOX2 in mESCs (*Gao et al., 2009*; *Lai et al., 2012*). In both these studies, differentiation of mESCs promotes the PARP1-SOX2 interaction. We found PARP1 interaction with SOX2 is disrupted by *O*-GlcNAc and decreased GlcNAc-S248 in differentiating ESCs, consistent with a model in which the developmentally regulated decrease in S248 *O*-GlcNAcylation promotes PARP1 interaction. During differentiation, the SOX2-PARP1 interaction inhibits SOX2 binding to enhancers of genes that are necessary for self-renewal (*Lai et al., 2012*). We find that SOX2$^{S248A}$ exhibits increased association with PARP1 without decreased occupancy of pluripotency genes under self-renewal conditions. This result indicates the SOX2-PARP1 interaction alone does not inhibit SOX2 binding to pluripotency gene targets and suggests additional developmentally regulated alterations in SOX2 PTMs or interaction partners contribute to control of SOX2 occupancy.

The increase in SOX2 S253 phosphorylation in fS248A-Tg mESCs suggests there is potential for crosstalk between *O*-GlcNAcylation and phosphorylation in the TAD peptide in mESCs. In general, phosphorylation and *O*-GlcNAcylation both occur in structurally flexible regions of proteins, although there is no correlation or anti-correlation in the linear proximity of one PTM to another (*Trinidad et al., 2012*). It is unlikely phospho-S253 has a substantial impact on pluripotency gene expression, as S253 phosphorylation is dispensable for mESC self-renewal (*Ouyang et al., 2015*). However, because the TAD of SOX2 is an unstructured domain (*Reményi et al., 2003*), and is *O*-GlcNAcylated and phosphorylated, the molecular basis for the potential crosstalk may be specific to a different cellular context.

*O*-GlcNAc signaling is essential for pluripotency. However, reliable *O*-GlcNAc site identification, as well as investigation into this PTM's function, is in their infancy. This study shows that although global *O*-GlcNAcylation is necessary for mESC self-renewal, a key pillar of pluripotency can be inhibited by *O*-GlcNAc modification. This work provides a new mechanism for the regulation of SOX2 through *O*-GlcNAcylation, and illustrates the role of this PTM in pluripotency and self-renewal is more complex than previously appreciated.

## Materials and methods

### Cell line derivation

2TS22C mESCs (accession number AES0125), the parental cell line for derivation of tagged SOX2 lines (*Masui et al., 2007*), were obtained through Riken BioResource Center. Identity of this line, which contains a SOX2 transgene under control of a tet-repressible promoter, was authenticated by culturing cells with and without doxycycline and examining SOX2 expression by Western blotting. To create the fSOX2-Tg ESC line, 4 ug of the plasmid CAG-3xF-Sox2 was transfected with Lipofectamine 2000 (Thermo Fisher [Invitrogen], Waltham, MA) into a 6-well plate containing 2TS22C cells.

2TS22C mouse ESCs express SOX2 under control of a tetracycline-repressible system. Twenty-four hours after transfection 1 ug/mL doxycycline was added to silence expression of the TetO Sox2. Forty-eight hours after transfection, 7 ug/mL puromycin was added, to select for the integration and expression of CAG-3xF-Sox2. After about two weeks, colonies exhibiting the typical ESC morphology were expanded and tested via western blot, morphology and alkaline phosphatase staining (Clontech, Mountain View, CA). The same strategy was used to generate the fS248A-Tg and haSOX2-Tg ESC lines.

## Cell culture

Mouse ESC lines were routinely passaged by standard methods in ESC media (KO-DMEM, 10% FBS, 2 mM glutamine, 1X non-essential amino acids, 1x pencillin/streptomycin, 0.1 mM b-mercaptoethanol and recombinant leukemia inhibitory factor). 2TS22C, fSOX2-Tg, fS248A-Tg, and haSOX2-Tg ESCs were cultured in ESC media with 1 ug/mL doxycycline hyclate (Sigma). KI mESCs (*Lai et al., 2012*) were cultured in N2B27 plus 2i (Thermo Fisher [Life Technologies], Waltham, MA) and LIF or 500 uM retinoic acid (Sigma-Aldrich, St.Louis, MO). Mycoplasma testing was carried out every two months until the cells were found to be negative for several successive tests.

## Plasmids

CAG-3xF-Sox2 was constructed by inserting mouse Sox2 cDNA into pCMV-3xFLAG 7.1 (Sigma-Aldrich) and then subcloning 3xF-Sox2 via InFusion cloning (Clonetech) into CAG-HA-Sox2-IP (Addgene plasmid 13459), replacing the HA tag with the triply FLAG tagged. S248A mutations in CAG-3xF-Sox2 were performed with Quickchange site directed mutagenesis (Agilent Technologies [Stratagene], Santa Clara, CA) (Supp Table S4). pMXs-mouse Sox2 wild type or S248A, along with pMXs-Oct4, Klf4 and c-Myc were used (*Okita et al., 2007*; *Takahashi and Yamanaka, 2006*). The pMXs-3xF-Sox2 was created by cloning Sox2 into pCMV-3xFLAG 7.1 (Sigma-Aldrich) and then subcloning 3xF-Sox2 via InFusion cloning (Clonetech) into pMXs.

## Antibodies and alkaline phosphatase

Antibodies were purchased from:

Abcam, United Kingdom: SOX2 ab75179 (immunofluorescence [IF] and in vitro assay WB, TUBULIN GTU-88 ab11316 (Western blot [WB]), HA ab13834 (WB, IP), OCT4 ab19857 (IF)

Reprocell USA INC, Boston, MA: NANOG, RCAB002P-F (IF)

SIGMA-Aldrich: OGT DM-17 (WB), FLAG A8592 (WB), FLAG F1804 (IF and ChIP)

Santa Cruz Biotechnology Inc, Santa Cruz, CA: OGT SC32921 (in vitro assay WB), OCT4 SC8628x (WB)

Bethyl Laboratories, Inc, Montgomery, TX: SMARCA4 A300-813A (WB), PARP1 A301-375A (WB), GATAD2B 301-281A (WB)

Pierce/ThermoFisher: O-GlcNAc MA1072 (WB)

ActiveMotif, Carlsbad, CA: PARP1 39,559 (in vitro assay WB)

Bio-Rad Laboratories, Inc, Hercules, CA: goat anti-rabbit HRP conjugate 172–1019 and goat anti-mouse HRP conjugate 172–1011 (WB)

Jackson Laboratory, Bar Harbor, ME: Alexa Fluor 488-donkey anti-rabbit IgG 711-545-152 (IF)

Alkaline phosphatase activity staining was performed according to manufacturer's instructions (Stemgent, Cambridge, MA, 00–0055).

## SOX2 purification for PTM characterization

KI mESCs, fSOX2-Tg or fS248A-Tg cells were expanded to one to three 15 $cm^2$ dishes depending on the experiment. Cells were harvested by trypsinization, washed once with cold PBS and frozen in liquid nitrogen. Whole cell pellets were lysed in RIPA buffer without SDS, containing 500 nM Thiamet G (Caymen Chemicals, Ann Arbor, MI), 1X HALT protease and phosphatase inhibitors (Thermo Fisher[Pierce]), 2 mM TCEP (Sigma) and 20 mM *N*-ethylmaleimide (Sigma-Aldrich) and sonicated (with a probe sonicator on methanol ice for three rounds of pulses, 3 s on, 2 off, 10 s total, at 35%). Anti-FLAG-based purifications were performed with anti-FLAG M2 Dynabeads (Sigma-Aldrich, M8823). Whole cell lysates were incubated with M2 beads at room temperature for 75 min, washed once with lysis buffer and three times with 25 mM ammonium bicarbonate (ABC) with 150 mM

NaCl. Proteins were eluted with 100 mM glycine pH 4 for five minutes. Western blot and SDS-PAGE analysis was used to assess purification efficiency. For HA-SOX2 purification, anti-HA antibodies were coupled to aldehyde-coated magnetic beads (BioClone, San Diego, CA) via reductive amination in 20 mM bicine pH 7.8 and purified as above. Two 10 cm$^2$ dishes of CD1 MEFs infected with OS$^{FLAG-WT}$KM or OS$^{FLAG-S248A}$KM for six days were analyzed the same way.

## Mass spectrometric analysis of SOX2 PTMs

Silver or Coomassie stained SDS-PAGE gel bands were excised and digested in-gel with sequence grade trypsin (Roche, Switzerland). After 5% formic acid/50% acetonitrile extraction, peptides were dried by vacuum centrifugation, gel particulates were removed via C18 Zip Tips (MERCK Millipore, Billercia, MA), dried, resuspended in 0.1% formic acid and analyzed by LC-MS/MS. Chromatography was performed on a Nanoacquity HPLC (Waters, Milford, MA) at 400 nl/min with a BEH130 C18 2.1x150 mm column (Waters). A 90- or 120-minute gradient from 98% solvent A (0.1% formic acid) to 22% solvent B (0.1% formic acid in acetonitrile) was used. Peptides were analyzed by an LTQ-Orbitrap Velos mass spectrometer (Thermo Scientific, Waltham, MA). After the survey scan of m/z 400–1,600 was measured in the Orbitrap at 30,000 resolution, the top three multiply charged ions were selected for both HCD and ETD. Automatic gain control for MS/MS was set to 2000. Normalized collision energy for HCD was set at 35 while the ETD activation time was charge state dependent, based on 100 ms for doubly charge precursors. Supplemental activation was implemented for ETD reactions. Dynamic exclusion of precursor selection was set for 25 s.

WT GlcNAc-S248, WT unmodified and the S248A SOX2 TAD were synthesized by New England Peptide and analyzed as described above.

## Data analysis

Fragment mass spectra were converted into peaklists using the in-house software PAVA. HCD and ETD data were searched separately using ProteinProspector version 5.10.0 against the UniProt database with a concatenated database. Only mouse and human genomes were used for the database searching. Precursor tolerance was set to 10 ppm, whereas fragment mass error tolerance was set to 0.6 Da for ETD and 20 ppm for HCD. N-terminal acetylation, methionine oxidation, loss of N-terminal methionine and glutamate conversion to pyroglutamate were allowed as variable modifications. For ETD data, HexNAc modifications to serine and threonine residues and phosphorylation to serine/threonine/tyrosine was allowed as variable mass modifications. For HCD, phosphorylation was searched the same way though HexNAc was considered as a neutral loss. Methylation (mono, di- and tri-) of K and R, monomethylation of D, E and H (artifact from MeOH fixing PAGE gels), acetylation of K and R, and ADP-ribosylation to C, E, K, N, S, and R were searched separately. SLIP scoring was used to distinguish possible positional isomers of HexNAc and/or phosphopeptides (*Baker et al., 2011*). Relative abundances of each modified or unmodified peptide were calculated using the ICIS area calculated from XICs in Xcalibur (Thermo Scientific) at a 10 ppm mass tolerance.

## Analysis of 3xF-SOX2 interactors

Six to ten 10 cm$^2$ plates worth of haSOX2-Tg, fSOX2-Tg or fS248A-Tg cells were harvested for nuclear extract preparations in biological duplicate. Nuclear extracts were as previously described with minor modifications (*Dignam et al., 1983*). Buffers A, C and D were supplemented with 2 μM Thiamet G, 2 μM PUGNAc (Tocris Bioscience, United Kingdom) and 1X HALT protease and phosphatase inhibitors and instead of dialysis of extracts, two volumes of buffer D were used to dilute salt concentration. Seven μL of M2 beads per 10 cm$^2$ plate were used per co-IP and samples were nutated at 4°C for two hours. Beads were washed once with Buffer D plus inhibitors, then twice with 50 mM ABC with 150 mM NaCl. Each wash was only as long as it took to transfer the beads to a new, cold tube and place on the magnetic rack. Beads were then resuspended in 50 μL 100 mM ABC with 500 ng trypsin and shaken at 37°C for one hour. Supernatant was transferred to a new tube, the beads were washed once with 50 μL ABC and combined to digest overnight. Digestions were desalted with one or two Zip Tips, depending on size of experiment, and dried via vacuum centrifugation. IP-WB experiments were performed similarly except proteins were eluted with 2X SDS-PAGE loading buffer without reducing agent.

Chromatography was performed on a Nanoacquity HPLC (Waters) at 400 nl/min with an EASY-spray 15 cm x 75 µm ID, PepMap C18, 3 µm column (Thermo Scientific). A 90-minute gradient from 98% solvent A (0.1% formic acid) to 22% solvent B (0.1% formic acid in acetonitrile) was used. Peptides were analyzed on a Q-Exactive Plus mass spectrometer (Thermo Scientific). After the survey scan of m/z 400–1,600 was measured in the Orbitrap at 70,000 resolution, the top ten multiply charged ions were selected for HCD and measured at 17,500 resolution. Normalized collision energy for HCD was set at 35, Dynamic exclusion of precursor selection was set for 25 s.

The label-free quantiation (LFQ) feature of MaxQuant (1.5.1.0) was used to quantify protein signals for proteins identified in the co-IP experiments. For SOX2, only non-TAD peptides were used for protein level quantitation. Proteins were determined to be SOX2 interactors by taking the average ratio of the LFQ intensity of the protein of interest (POI) from the FLAG-tagged mESC lines over the HA-tagged mEScs. This average ratio was $log_2$-transformed, and was normalized by the global median. If the POI was two standard deviations from the mean it was considered specific to SOX2. Most proteins discussed were manually verified using Skyline.

To determine POIs that differentially interact with wild type or S248A SOX2, we took the ratio of the $log_2$-transformed ratio of FLAG/HA for S248A over the wild type. After normalization by the global median, POIs were considered to be differential interactors if they had a z-score of greater than 1.5. All of these peptides were manually verified using Skyline.

For targeted analyses, FLAG-tagged MBD3 mESCs, a generous gift from the Fazzio laboratory (*Yildirim et al., 2011*) were analyzed as described above. From these analyses the top two most intense, non-homologous peptides identified from NuRD subunits were used to monitor their co-purification with SOX2 isoforms in fSOX2-Tg and fS248A-Tg cells in three separate, biological replicates. XICs were extracted manually using Xcalibur software. Relative enrichment of proteins co-purified with either SOX2 form was determined as described above. Peptides used for this analysis are listed in Supplementary Table 1b.

Samples for co-IP Western blot corroboration of LC-MS data were performed as described above in biological duplicate. Antibodies used are described above.

## Reprogramming experiments

MEFs were derived from wild type CD1 mice or the Nanog-Gfp-IRES-Puror mice (*Okita et al., 2007*) and cultured in MEF media (KO-DMEM, 10% FBS, 2 mM glutamine, 1X non-essential amino acids, 1X pencillin/streptomycin, and 0.1 mM β-mercaptoethanol). pMXs vectors containing *Oct4, c-Myc, Klf4, eGfp, dsRed*, or wild type or S248A *Sox2*, with or without FLAG, were transfected with Fugene 6 (Promega, Madison, WI, E2691) into PlatE cells. Twenty-four hours after transfection, the media was changed. The next day, the retroviral supernatant was collected from transfected PlatE cells, filtered through 0.4 um filters and combined with each other at equal ratios. Polybrene (Merck Millipore, TR-1003-G) was added to a final concentration of 4 ug/ml. The virus-containing media was added to *Nanog-Gfp* or wild type MEFs that were passaged less than five times. Media was replaced the next day and every other until six days after transduction. At day six, MEFs were trypsinized and either prepared for experiments or 1000 cells were plated onto γ-irradiated SNL feeders. These 1000 MEFs were cultured in ESC media until GFP+ colonies were counted at day 20 (*Nanog-Gfp* MEFs).

Microinjection of iPSCs to generate chimera mice was conducted at Cornell University Stem Cell and transgenic core facility. iPSCs were grown on mouse embryonic fibroblasts (produced at the Cornell stem cell core) and mitotically inactivated by irradiation (3000 Rads). To produce donor embryos, wild type albino mice of the strain http://jaxmice.jax.org/strain/000058.html were mated, embryos were flushed from the uterus at day 3.5, and the iPSCs were injected into the blastocyst of each embryo (15–30 cells per embryo). Injected embryos were then transferred to 2.5-day pseudo pregnant recipient animals and pup chimaerism was determined by coat color. Chimeras were mated to age-matched wild type animals of the same albino strain used for embryo donors. iPSC contribution to the germline was determined by coat color of the resultant pups.

## Microarray analysis

Total RNA was extracted with Trizol (Thermo Fisher [Invitrogen]) according to manufacturer's instructions. Arraystar Inc, Rockville, MD(http://www.arraystar.com) prepped and hybridized the samples, and performed the data analysis.

For RT-qPCR, 1 μg total RNA was reverse transcribed to cDNA with iScript (Bio-Rad Laboratories), diluted 1:20 or 1:50, depending on the abundance of the transcript, and 4 μL was used. Quantitative PCR was performed on a CFX Connect Real-time PCR detection system (Bio-Rad laboratories) with SensiFast SYBR Lo-ROX PCR master mix (Bioline, Taunton, MA, BIO-94020). Fold enrichment was determined by $2^{-(\Delta Cq)}$ method ($\Delta Cq = Cq(gene)-Cq(GusB)$). Primers are listed in *Supplementary file 1d*.

## Recombinant SOX2-OGT reaction and GST-PARP1 pulldown

The recombinant poly-His tagged human OGT (His-OGT) expression plasmid was a generous gift from Suzanne Walker. His-OGT was expressed as previously described (*Gross et al., 2005*). His-OGT was purified by Ni-NTA agarose resin (Qiagen, Germany), eluted and buffer exchanged into 50 mM Tris-HCl, pH 7.8, 300 mM NaCl. To generate a biotinylated SOX2 (Bio-SOX2), the human SOX2 cDNA was cloned into the expression vector, pGV358avi, as a fusion construct linked to an N-terminal Avi-tag and a C-terminal intein-chitin binding domain (*Redding et al., 2015*). Bio-SOX2 was produced in E.coli BL21(DE3) in media supplemented with 200 nM biotin. The protein was purified from a clarified lysate by passage over a chitin column (New England Biolabs (NEB), Ipswich,MA) and incubated overnight in buffer containing 20 mM Tris-HCl, pH 8.5, 500 mM NaCl, 1 mM EDTA, 10% glycerol and 50 mM DTT. Bio-SOX2 was then eluted from the chitin column and dialyzed into storage buffer (20 mM Tris-HCl,pH 8.5, 500 mM NaCl, 1 mM EDTA, 10% glycerol and 1 mM DTT) for long term storage at -80°C.

Equimolar ratios of Bio-SOX2 and His-OGT were incubated in an *O*-GlcNAc assay buffer (50 mM Tris pH 7.4, 12.5 mM MgCl$_2$, 2% glycerol, 0.2 mM PMSF, 1 mM DTT) for one hour at 37°C with or without the sugar donor, UDP-GlcNAc (100 μM). After one hour, 50 μl of streptavidin magnetic beads (Thermo Fisher, 11205D) were resuspended in PBS/BSA(0.1%) buffer were added and the reactions were incubated for 45 min at 4°C. Bio-SOX2 bound beads were washed twice with PBS/BSA, once with low salt buffer (PBS/BSA+150 mM NaCl) and once with high salt buffer (PBS/BSA +300 mM NaCl) to eliminate His-OGT and unincorporated UDP-GlcNAc. Finally, the beads were resuspended in 50 μl of PBS/BSA buffer and incubated at 4°C for 1 hr with recombinant GST-PARP1 protein (0.14 μM) (Sigma-Aldrich, SRP0192). The beads were washed three times in PBS/BSA buffer and eluted in 1X SDS-PAGE loading buffer. Pull down and input samples were resolved on a 10% SDS-PAGE gel and Western blotted for SOX2, (Abcam, ab75179), OGT (SantaCruz Biotechnology Inc,SC 32921) *O*-GlcNAc (Thermo Fisher [Pierce, MA1072) and PARP1 (Active Motif, AM 39559).

## Knockdown

siRNA pools (esiRNAs) were created by in vitro cleavage of double-stranded RNA (*Fazzio et al., 2008*). In vitro transcription templates for double stranded RNAs were generated using the primers acquired from the Riddle database (*Kittler et al., 2007*). Transfections were performed using Lipofectamine RNAiMax according to manufacturer's instructions.

## ChIP sample preparation

The cells from ~80% confluent 10 cm$^2$ dishes were crosslinked with 1% formaldehyde for 10 min at room temperature. After quenching with 125 mM glycine for 5 min, the cells were washed twice with ice cold PBS. Cell pellets were resuspended in 1ml lysis buffer 1 (50 mM HEPES-KOH, pH 7.6,140 mM NaCl,1 mM EDTA, 10% (v/v) Glycerol, 0.5% NP-40,0.25% Triton X-100, complete protease inhibitor cocktail (PIC) (Roche,11697498001) for 10 min at 4°C, followed by centrifugation at 1,350 x g for 5 min at 4°C. Discard the supernatant and resuspend the pellet in 1 ml of lysis buffer 2 (10 mM Tris-HCl pH 8.0, 200 mM NaCl, 1 mM EDTA, 0.5 mM EGTA, PIC) for 10 min in at 4°C, followed by centrifugation at 1,350 x g for 5 min at 4 °C. Finally resuspend the cell pellet in 700 μl of lysis buffer 3 (10 mM Tris-HCl pH 8.0, 100 mM NaCl, 1 mM EDTA, 0.5 mM EGTA, 0.1% sodium deoxycholate, 0.5% N-Lauroylsarcosine, 1X PIC) and sonicate to desired length using Bioruptor

(UCD-200; Diagenode, Denville, NJ) for 30 min [10 min x 3 times (each cycle consists of 30 s on, 30 s off) at high settings]. Clarify the lysates by centrifugation at 10,000 rpm in a microcentrifuge for 10 min at 4°C. Transfer the sonicated chromatin to a new tube and store at -80°C or use it immediately for chromatin immunoprecipitation (ChIP). We used two sets of chromatin (long and short) for ChIP and library preparation. Long ChIP samples were generated using sonicated chromatin consisting of DNA fragments ranging from sizes 1 kb–500 bp. Short ChIP samples were generated using sonicated chromatin consisting of DNA fragments ranging from sizes 500–200 bp.

First, FLAG-coupled protein G beads were generated by incubating 10 ug of FLAG antibody (Sigma-Aldrich, F1804) with 50 µl of prewashed protein G magnetic beads (NEB S1430S) in PBS/BSA (5 mg/ml) buffer at 4°C, overnight. Chromatin Immunoprecipitation is performed by incubating these FLAG-coupled protein G beads with the sonicated chromatin in ChIP buffer (20 mM Tris-HCl pH 8.0, 150 mM NaCl, 2 mM EDTA, 1% Triton X-100) at 4°C, overnight. 10% of the total chromatin used for the ChIP is set aside as input sample and stored at -20°C until further use. The magnetic beads were washed twice with ChIP buffer, once with ChIP buffer containing 500 mM NaCl, four times with RIPA buffer (10 mM Tris-HCl pH 8.0, 0.25 M LiCl, 1 mM EDTA, 0.5% NP-40, 0.5% sodium deoxycholate), and once with TE buffer (10 mM Tris-HCl,pH 8.0, 1 mM EDTA). Finally elute DNA from the beads by adding 100 µl of elution buffer (20 mM Tris-HCl pH 8.0, 100 mM NaCl, 20 mM EDTA, 1% SDS) twice and incubating for 15 min at 65°C. The eluted DNA and the input samples were then reverse crosslinked at 65°C for overnight, followed by RNase A(0.2 mg/ml) digestion at 37°C for 2 h and Proteinase K (0.2 mg/ml) (NEB, P8102S) digestion at 55°C for 1 hr. The ChIP and input DNA were recovered by phenol-chloroform extraction and ethanol precipitation.

## ChIP-seq Library preparation

ChIP samples were end-repaired, A-tailed and adaptor ligated using barcode adaptors. Briefly, DNA was end-repaired using a combination of T4 DNA polymerase, E. coli DNA Pol I large fragment (Klenow polymerase) and T4 polynucleotide kinase. The blunt, phosphorylated ends were treated with Klenow fragment (exo minus) and dATP to yield a protruding 3- 'A' base for ligation of barcoded adapters which have a single 'T' base overhang at the 3' end. DNA purification on Qiagen mini elute columns was performed following each enzyme reactions. The adaptor ligated material was then PCR amplified with Phusion polymerase using 16 cycles of PCR before size selection of 200–350 bp fragments on a 2% agarose gel. The ChIP-seq library samples were purified using Qiagen gel extraction kit, and its concentration was determined on Agilent Bioanalyzer using High Sensitivity DNA chip (Agilent Technologies, Santa Clara, CA). Libraries with different barcodes were multiplexed together at equimolar concentrations and single-end sequencing (50 bp) was performed at Center for Advanced Technology, genomics core facility, UCSF. Each lane of the HiSeq 2000 (Illumina, San Diego, CA) had five libraries (four ChIP DNA and one input DNA) multiplexed with barcodes added to the 5' end of the sequence.

## ChIP-seq data analysis

Reads were identified and then mapped to the mm9 assembly of the mouse genome using the Bowtie aligner (*Langmead and Salzberg, 2012*). Normalized and background-corrected measures of ChIP signal were created by randomly choosing 10 million unique tags from each dataset, calculating the tag density within 75 bp of each 20 bp bin of the mm9 assembly mouse genome, and then subtracting the matched 10-million-tag-normalized input tag density from each dataset. UCSC genome browser was used to visualize the ChIPseq peaks.

All ChIP-seq peaks present in both the long and short biological replicates were used for motif analysis. Motifs were identified by running MEME (*Bailey and Elkan, 1994*) on fS248A-Tg mESC peaks (-dna -mod zoops -nmotifs 3 -minw 6 -maxw 30 -time 6058 -revcomp -maxsize 1000000). De novo motifs were matched to known motifs using Tomtom (*Gupta et al., 2007*).

The ChIP-seq and microarray data discussed in this publication have been deposited in NCBI's Gene Expression Omnibus and are accessible through GEO Series accession number GSE69594 (http://www.ncbi.nlm.nih.gov/geo/query/acc.cgi?acc=GSE 69594).

## Relative quantitation of AP staining

Cells were stained for AP activity as previously described above. NIS-Elements Basic Research software automatically acquired the darkness of the staining. Four fields of view for each experiment was taken and averaged to plot the change in darkness between gene of interest knockdown and *Gfp*.

## Acknowledgements

We greatly appreciate the generous gifts from Tom Fazzio for the fMBD3 mESCs, Tim Townes for the KI SOX2 mESCs, Suzanne Walker for the His-OGT plasmid, Sy Redding, and David Bauer for Avi-tagged human Bio-SOX2. We thank Anne Claude Gingras, Joe Kleigman, Michael Lopez, Katie Worringer, and Anthony Shiver for technical assistance. Thank you to Kathyrn Lovero and Jonathan Fistorino for useful discussion, and Emily Myers and Lucas Sullivan for critical assessment of this manuscript. We also thank the Arraystar and the Gladstone Bioinformatics core for assistance analyzing the microarray and ChIP-seq data, respectively. This work was supported by the Biomedical Technology Research Centers program of the NIH National Institute of General Medical Sciences, NIH NIGMS P41GM103481, 1S10RR019934, Howard Hughes Medical Institute (purchase of ETD mass spectrometer) and the Dr. Miriam and Sheldon G. Adelson Medical Research Foundation (ALB), NIH R01GM085186, CIRM RB4-05990, and University of California San Francisco Program for Breakthrough Biomedical Research (BP). SAM is supported by a National Institutes of Health National Institute of General Medical Sciences T32 training grant, the Genentech Predoctoral Fellowship Program and the QBC Fellowship for Interdisciplinary Research. The Cornell Stem Cell Core is supported by the Empire State Stem Cell fund through NYSDOH Contract # C024174, and opinions expressed here are solely those of the author and do not necessarily reflect those of the Empire State Stem Cell Fund, the NYSDOH, or the State of NY.

## Additional information

### Competing interests

SY: Scientific advisor of iPS Academia Japan without salary. The other authors declare that no competing interests exist.

### Funding

| Funder | Grant reference number | Author |
|---|---|---|
| National Institute of General Medical Sciences | NIGMS P41GM103481 | Alma L Burlingame |
| Howard Hughes Medical Institute | | Alma L Burlingame |
| California Institute for Regenerative Medicine | CIRM RB4-05990 | Sailaja Peddada<br>Nilanjana Chatterjee<br>Barbara Panning |
| Dr. Miriam and Sheldon G. Adelson Medical Research Foundation | | Samuel A Myers |
| University of California, San Francisco | | Samuel A Myers<br>Sailaja Peddada<br>Nilanjana Chatterjee<br>Tara Freidreich<br>Kiichrio Tomoda<br>Gregor Krings<br>Sean Thomas<br>Michael Broeker<br>Matthew Thomson<br>Katherine Pollard<br>Barbara Panning |

The funders had no role in study design, data collection and interpretation, or the decision to submit the work for publication.

## Author contributions
SAM, SP, KT, GK, Conception and design, Acquisition of data, Analysis and interpretation of data, Drafting or revising the article; NC, MB, Acquisition of data, Analysis and interpretation of data; TF, ST, KP, SY, Analysis and interpretation of data, Drafting or revising the article; JM, Acquisition of data, Drafting or revising the article; MT, Conception and design, Drafting or revising the article; ALB, BP, Conception and design, Analysis and interpretation of data, Drafting or revising the article

## Author ORCIDs
Barbara Panning, [ID] http://orcid.org/0000-0002-8301-1172

# Additional files

## Supplementary files
• Supplementary file 1. (a) Differentially expressed genes between fSOX2-Tg and fS248A-Tg cells determined by microarray. P-values, absolute fold change and direction of regulation are listed in columns 2, 3, and 4, respectively. (b) Proteins, listed by Gene name, found to interact with SOX2 in mESCs in this study. *FLAG/HA IP* (column 1) lists proteins found to specifically interact with SOX2 in either fSOX2-Tg or fS248A-Tg cells but not in haSOX2-Tg cells (FLAG IP control). *fS48A-Tg Int./fSOX2-Tg Int.* (column 2) indicates the log2 fold-enrichment of the SOX2-protein interaction between the two cell lines. *Z-score (fS48A-Tg/fSOX2-Tg)* indicates the enrichment score for a SOX2-protein interaction between the two cell lines. Previously identified SOX2 interactors are listed for the associated references. Full references can be found in the main text. (c) Peptides sequences and m/z values used for the targeted analysis described in *Figure 6c*. (d) Primers used in this study.

## Major datasets
The following datasets were generated:

| Author(s) | Year | Dataset title | Dataset URL | Database, license, and accessibility information |
|---|---|---|---|---|
| Panning B | 2016 | SOX2 O-GlcNAcylation alters its protein-protein interactions and genomic occupancy to modulate gene expression in pluripotent cells | http://www.ncbi.nlm.nih.gov/geo/query/acc.cgi?acc=GSE69594 | Publicly available at the NCBI Gene Expression Omnibus (Accession no: GSE69594). |
| Panning B, Peddada S | 2016 | SOX2 O-GlcNAcylation alters its protein-protein interactions and genomic occupancy to modulate gene expression in pluripotent cells [ChIP-seq] | http://www.ncbi.nlm.nih.gov/geo/query/acc.cgi?acc=GSE69592 | Publicly available at the NCBI Gene Expression Omnibus (Accession no: GSE69592). |
| Panning B, Peddada S, Myers SA, Krings G | 2016 | SOX2 O-GlcNAcylation alters its protein-protein interactions and genomic occupancy to modulate gene expression in pluripotent cells [array] | http://www.ncbi.nlm.nih.gov/geo/query/acc.cgi?acc=GSE69593 | Publicly available at the NCBI Gene Expression Omnibus (Accession no: GSE69593). |

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
