## [Decision Letter]

Thank you for submitting your work entitled "O-GlcNAc regulates SOX2 protein-protein interactions and genomic occupancy to control pluripotency gene expression" for peer review at *eLife*. Your submission has been favorably evaluated by three reviewers, and the evaluation was overseen by a Reviewing editor and Janet Rossant as Senior editor.

The reviewers have discussed the reviews with one another and the Reviewing editor has drafted this decision to help you prepare a revised submission.

Summary:

The manuscript is a very interesting study regarding the role of SOX2-S248 *O*-GlcNAcylation in the context of ESC differentiation and reprograming. The authors use several functional and genome/proteome-wide assays to investigate the role of *O-*GlcNac in pluripotency and reprogramming by focusing on the transcription factor SOX2.

Myers et al. follow up on their previous observation that the pluripotency factor SOX2 is *O-*GlcNac-modified at Ser-248 in mouse ESCs. Here, they show that *O-*GlcNac-modified SOX2 decreases with differentiation and increases during reprogramming into iPSCs. Ectopic expression of a mutant form of SOX2 that cannot be *O-*GlcNac-modified (S248A) in ESCs supports self-renewal, gives rise to less differentiated colonies and suppresses differentiation. Similarly, ectopic expression of S248A in combination with Oct4, Klf4and c-Myc gives rise to more iPSC colonies than OCT4, SOX2 (WT), Klf4 and c-Myc expression. The authors finally show that ESCs expressing SOX2 S248A form associate with ca. 1,000 loci that are different from WT SOX2 in ESCs and lack nearby OCT4 binding sites. Moreover, proteomic analysis suggests that the S248A form of SOX2 associates less efficiently with PARP1 and other proteins, thus providing a partial explanation for the differentiation defect of mutant ESCs. This study makes a number of interesting observations on a modification that has received a lot of attention in the stem cell field over the last few years. This is an interesting area and this approach is an important one. The manuscript should be acceptable for publication once the authors eliminate the weaknesses outlined below. Various concerns can be addressed by editing text and labels, and by clarifying experimental details in particular with respect to statistics/quantification/replicates. In addition, there is a need for a few additional experiments to confirm the validity of some of the main findings and conclusions, which are perceived by all reviewers as relatively straightforward for the authors.

Essential revisions:

1) Extracted ion chromatograms are sub-optimal to show quantitative values for peptide abundance since ionization can vary from run to run (i.e. comparing Figure 1 to 1C). More developed quantitative methods would be much more informative (SILAC, isobaric tagging, etc.). The authors could normalize the area under a peak for a given SOX2 isoform to the area under all SOX2 isoforms to establish relative quantitation; however, the authors would also need to prove that there is no difference in ionization efficiency between WT SOX2 and S248A SOX2. The authors could also use an orthogonal technique such as Western blot if a modification specific antibody is available.

2) The retention times for the *O-*GlcNAc modified fSOX2-Tg vs. fS248A-Tg is different: is that to be expected for a single amino acid change? Moreover, the authors mention that the asTAD peptide could be modified at T258 with *O-*GlcNAc but do not examine the potential role further. Is this the reason for the residual peak corresponding to an *O-*GlcNAc modified peptide in all of the traces? (i.e., with the S248A cells, one would expect no blue trace otherwise). Importantly, the authors need to address at some level why T258 was not further explored – they did not make this explicit.

3) Figure 2—figure supplement 1: the authors make conclusions based on this figure, but they do not show statistics. If there is statistical significance associated with the blue and red bars that should be clearly depicted. Moreover, how many times were the experiments in Figure 2—figure supplement 1 performed? It is important to convince the reader and the reviewers that these results are reproducible and give some statistics in the paper.

4) In the reprogramming experiments, the authors need to show that the SOX2 WT and S248A vectors are expressed at comparable levels. If they are not, this would be a much more trivial explanation for the observed phenotype.

5) The authors need to better describe and characterize the SOX2-deficient ESC lines (originally made by Masui et al.) harboring SOX2 overexpression constructs, and need to provide more details on how the cell line fSOX2-Tg and F248A-Tg have been created. According to the material method, 2TS22C cells have been stably transfected with CAG-3xF-Sox2 WT or S248A. One reviewer was wondering why the endogenous SOX2 present in the parental line visible on the top WB in Figure 3 is not expressed in fSOX2-Tg and f248A-Tg? Therefore, the parent cell line setup needs to be explained. The authors should also analyze expression of a few pluripotency markers such as OCT4, SOX2 and NANOG by IF. AP staining is not a quantitative method and should be avoided (regardless, Figure 3: AP staining difficult to see).

6) In Figure 3, fSOX2 WT and fSOX2-248A exhibit similar cell localization but localization of endogenous SOX2 in parental cell line should be shown as well to appreciate the physiological localization of the overexpress isoforms of fSOX.

7) The authors claim that many genes associated with pluripotency are increased in SOX2 S248A mutants as judged by microarrays. It would be helpful to provide GSEA to show that there is an actual enrichment.

8) Differences in SOX2 occupancy between SOX2 WT and S248 expressing ESCs are rather minimal (ca. 1,000 sites), which could reflect ChIP-to-ChIP variability. It is unclear whether these analyses were done in replicates. The authors need to convincingly show that these differences are real, for example by confirming differentially bound targets by ChIP-PCR using independent cell preparations. Alternatively, the authors could generate ChIP-seq data from biological replicates.

9) The authors suggest that the SOX2 S248 mutation directs SOX2 to sites that are not co-occupied by OCT4 based on the lack of a canonical OCT4 consensus sequence +/- 20 bp. This would be more convincing if the authors performed ChIP for OCT4 in the SOX2 S248 cells and showed that it does not co-occupy the sites.

10) Differences in protein-protein binding between SOX2 WT and S248A cells also need to be validated in a more quantitative manner. It is difficult to ascertain how robust the differences presented in Figure 6 really are. Has the MS and validation experiment been performed on the same sample or in biological replicates? In the second paragraph of the subsection “SOX2^S248A^ has altered protein-protein interactions”, the authors mention a 4-fold increase in binding partners but give no indication of how this quantitative value was calculated. Moreover, Figure 6 is a nice experiment to show that GlcNAcylation of SOX2 impairs its interaction with PARP1. However, loadings of SOX2 are not equal throughout the lane and the controls are missing. The authors should show the rate of SOX2 GlcNAcylation that is achieved in here. Also, it is obvious that OGT will interact with SOX2 but this interaction itself may impact the interaction of SOX2 with PARP1. A good control in here would be to incubated SOX2, PARP1 and OGT in presence or not of UDP-GLCNAc.

11) There are a number of typos that are apparent in the figures as well as throughout the text (e.g., Figure 1
*O-*GlcNAc – the "O" is a box; Figure 1, there is no top number on the y-axis, Figure 1 text: μM is μm rather than the "micro", F1a the double *O-*GlcNAc peptide is mislabeled. Figure 4—figure supplement 1, pictures are shifted… It also appears that tables are missing?

12) Corresponding mass-spectra of the data presented in Figure 2, Figure 2, Figure 3, Figure 3 are missing and should be provided.

13) Figure 1/C, 2B/C, 3C/D: the x-axis is labeled with RT, but there is no indication of the values associated with the retention time. This does not give the reader a good indication of how narrow a window the authors are examining. Moreover, Figure 1 appear to be stretched. Why is there a y ion annotated in Figure 1—figure supplement 4 and Figure 1—figure supplement 5? Is this from a secondary reaction and if so, why are they annotated? Other contaminating peaks that are much larger remain un-annotated.

14) *O-*GlcNAc modification of OCT4 has also been shown to be required for maintaining pluripotency. [Cell Stem Cell 17 May 2012 (doi: 10.1016/j.stem.2012.03.001)] While the authors reference this paper, they do not do so in the context of their OCT4 discussion and this is an important addition for the reader to best understand the state of the field.

15) The paper is solid but needs to be rewritten in parts to reflect what was actually tested and verified. Certainly SOX2 plays a key role in the maintenance of pluripotency and *O-*GlcNAc influences its function. A suggestion for the title may be: *O-*GlcNAc influences the protein-protein interactions mediated by the pluripotency factor SOX2. It is important that the authors do not leave the impression that this is a unique or sole means by which *O-*GlcNAc influences this important biological process. Moreover, it is important to be concise – for instance what does 'more pluripotent' stand for? Thus, the manuscript and the figures should be edited carefully throughout.

16) As the authors say, the phenotype of the S248A overexpression mutant is counterintuitive given the requirement for OGT in ESCs. I do not think that different contexts can explain this discrepancy, as the authors suggest, since both studies examine ESC self-renewal. It is more likely that the OGT KO phenotype is due to other *O-*GlcNac-modified proteins that have opposing effects to SOX2.

---

## [Author Response]

*1) Extracted ion chromatograms are sub-optimal to show quantitative values for peptide abundance since ionization can vary from run to run (i.e. comparing Figure 1 to 1C). More developed quantitative methods would be much more informative (SILAC, isobaric tagging, etc.). The authors could normalize the area under a peak for a given SOX2 isoform to the area under all SOX2 isoforms to establish relative quantitation; however, the authors would also need to prove that there is no difference in ionization efficiency between WT SOX2 and S248A SOX2. The authors could also use an orthogonal technique such as Western blot if a modification specific antibody is available.*

With modern high mass accuracy and high resolution mass spectrometers, coupled with high performance chromatography label free quantitation (LFQ) has become a powerful method of quantitation. This is especially true for less complex mixtures, such as co-immunoprecipitations or single proteins. We chose LFQ for this study because a majority of our analyses are not comparing isobaric peptides, the wild-type vs. the S248A peptides, for example. However, your point about the ionization efficiencies is very important. We have now included our LC-MS analysis of wild-type and S248A TAD synthetic peptides, as well the GlcNAc-S248 TAD synthetic peptide (Figure 3—figure supplement 2). These analyses show that the ionization efficiency between all the synthetic peptides are roughly equal.

*2) The retention times for the O-GlcNAc modified fSOX2-Tg vs. fS248A-Tg is different: is that to be expected for a single amino acid change? Moreover, the authors mention that the asTAD peptide could be modified at T258 with O-GlcNAc but do not examine the potential role further. Is this the reason for the residual peak corresponding to a O-GlcNAc modified peptide in all of the traces? (i.e., with the S248A cells, one would expect no blue trace otherwise). Importantly, the authors need to address at some level why T258 was not further explored – they did not make this explicit.*

The change in retention time between the wild-type and S248A peptides is expected. Removal of the hydroxyl group by changing the serine to alanine creates a less hydrophilic peptide. Our synthetic peptide analyses recapitulate the change in retention times between wild-type and mutant peptides.

The role of the other *O-*GlcNAc sites, while potentially interesting, was not the focus of this study. We did not examine the function of the T258 and S259 sites because they had already been studied by Jang et al. (Cell Stem Cell, 2012, Figure S4G). Jang et al. used an *O-*GlcNAc site prediction algorithm that identified T258 and S259 as potential sites. They performed reprogramming analysis with the double T258 and S259 mutant and showed a decrease in efficiency, while mutation of S248+T258+S259 exhibited normal efficiency. The loss of function phenotype of the T258+S259 mutant could be a result of impacting *O-*GlcNAcylation or perturbing protein structure. Since we had a gain of function phenotype with S248 alone and the Jang et al. triple mutant data also suggested a gain of function, we instead chose to focus on this modification site.

*3) Figure 2—figure supplement 1: the authors make conclusions based on this figure, but they do not show statistics. If there is statistical significance associated with the blue and red bars that should be clearly depicted. Moreover, how many times were the experiments in Figure 2—figure supplement 1 performed? It is important to convince the reader and the reviewers that these results are reproducible and give some statistics in the paper.*

We now include statistics and the number of replicates in the figure supplement.

*4) In the reprogramming experiments, the authors need to show that the SOX2 WT and S248A vectors are expressed at comparable levels. If they are not, this would be a much more trivial explanation for the observed phenotype.*

We now include Western blotting and immunostaining data showing that SOX2 and S248A are expressed at comparable levels at day 6 of reprogramming (Figure 2 and Figure 2—figure supplement 2).

*5) The authors need to better describe and characterize the Sox2-deficient ESC lines (originally made by Masui et al.) harboring Sox2 overexpression constructs, and need to provide more details on how the cell line fSOX2-TG and F248A-Tg have been created. According to the material method, 2TS22C cells have been stably transfected with CAG-3xF-SOX2^WT^ or S248A. One reviewer was wondering why the endogenous SOX2 present in the parental line visible on the top WB in Figure 3 is not expressed in fSOX2-Tg and f248A-Tg? Therefore, the parent cell line setup needs to be explained. The authors should also analyze expression of a few pluripotency markers such as OCT4, SOX2 and NANOG by IF. AP staining is not a quantitative method and should be avoided (regardless, Figure 3: AP staining difficult to see).*

The transgenic line originally made by Masui et al. has both endogenous copies of *Sox2* deleted, and a transgenic copy of the *Sox2* cDNA driven by tetracycline-repressible promoter is the sole source SOX2 (Masui et al. Nat Cell Bio, 2007). For our studies we introduced CAG-3xF-SOX2^WT^ or S248A under tet-repressed conditions, selecting for lines with levels of FLAG-tagged SOX2/S248A comparable to the amount of SOX2 in the original parental line (before repression). All experiments are carried out with the tet-SOX2 transgene repressed, which is why untagged SOX2 is no detectable in the blots in Figure 3. In the revised manuscript we provide a more detailed description of the generation of our transgenic ESC lines and include a new supplemental figure explaining the method (Figure 3—figure supplement 1). In addition, we now include Western blotting/IF for OCT4 and NANOG, showing that these are not substantially different between 3xF-SOX2^WT^ and 3xF-S248A mESCs.

We provide the NANOG, SOX2, and OCT4 immunostaining experiments requested by the reviewer, showing that the wild-type and mutant are not notably different for expression of these pluripotency transcription factors. We have also reduced the intensity of the AP staining to better show colony morphology.

*6) In Figure 3, fSOX2 WT and fSOX2-248A exhibit similar cell localization but localization of endogenous SOX2 in parental cell line should be shown as well to appreciate the physiological localization of the overexpress isoforms of fSOX.*

We apologize that the writing in our initial submission was not clear enough about how the ESC lines were generated. The 3xF-SOX2^WT^ and 3xF-S248A mESCs express SOX2 at levels comparable to the parental line by Western blotting (Figure 3) and exhibit similar localization (new panel in revised Figure 3)

*7) The authors claim that many genes associated with pluripotency are increased in SOX2 S248A mutants as judged by microarrays. It would be helpful to provide GSEA to show that there is an actual enrichment.*

We performed GSEA to examine whether self-renewal and/or differentiation genes exhibited preferential alteration in gene expression, and the analyses do not indicate that these pathways are substantially impacted by the S248A mutation. We have reworded the text to reflect this new analysis.

*8) Differences in SOX2 occupancy between SOX2 WT and S248 expressing ESCs are rather minimal (ca. 1,000 sites), which could reflect ChIP-to-O-ChIP variability. It is unclear whether these analyses were done in replicates. The authors need to convincingly show that these differences are real, for example by confirming differentially bound targets by ChIP-PCR using independent cell preparations. Alternatively, the authors could generate ChIP-seq data from biological replicates.*

We apologize that we were not clear enough in our original submission. In the Methods section we indicate that we performed two biological replicates (each with technical duplicates, for a total of 4 replicates) of the ChIP-seq. We now also state that replicates were performed in the figure legend accompanying the text. The files are deposited in GEO and available to the reviewers. Thus the 1000 new sites occupied in mutant relative to wild type (representing a 25% increase in the number of sites) are not likely to be due to ChIP-to ChIP variability.

*9) The authors suggest that the SOX2 S248 mutation directs SOX2 to sites that are not co-occupied by OCT4 based on the lack of a canonical OCT4 consensus sequence +/- 20 bp. This would be more convincing if the authors performed ChIP for OCT4 in the SOX2 S248 cells and showed that it does not co-occupy the sites.*

OCT4 ChIP followed by qPCR at S248A-specific peaks would certainly be informative about whether there is OCT4 binding in the vicinity of these new SOX2 binding sites. ChIP-qPCR doesn't provide the type of resolution necessary to indicate where the binding motif lies within the amplicon (in contrast to ChIP-seq, in which read counts indicate SOX2 and OCT4 peaks abut each other in the OCT4-SOX2 motif found in many active promoters in ESCs). Indeed, if we query for OCT4 motifs +/- 100 bp, a substantial fraction of S248A specific sites contain a predicted OCT4 binding site. Therefore, to address whether OCT4 is bound nearby SOX2 at the S248A-specific sites, ChIP-seq will be necessary. Performing such a ChIP-seq experiment is outside the scope of this manuscript, but is certainly something we will pursue in the future. For the resubmission we have changed the presentation of the analysis of ChIP-seq data. Instead of restricting our query to +/-20 bp of the SOX2 peaks, we used de novo motif identification to determine in an unbiased way what motifs predominant in the SOX2 peaks shared by wild type and S248A and in the S248A-specific peaks. Using this method, the canonical SOX2 motif is identified in the S248A-specific peaks and the canonical OCT4:SOX2 motif is identified in the peaks shared between wild type and mutant. This presentation more clearly highlights that mutant SOX2 site specificity is different from that of wild type SOX2.

*10) Differences in protein-protein binding between SOX2 WT and S248A cells also need to be validated in a more quantitative manner. It is difficult to ascertain how robust the differences presented in Figure 6 really are. Has the MS and validation experiment been performed on the same sample or in biological replicates? In the second paragraph of the subsection “SOX2^S248A^ has altered protein-protein interactions”, the authors mention a 4-fold increase in binding partners but give no indication of how this quantitative value was calculated. Moreover, Figure 6 is a nice experiment to show that GlcNAcylation of SOX2 impairs its interaction with PARP1. However, loadings of SOX2 are not equal throughout the lane and the controls are missing. The authors should show the rate of SOX2 GlcNAcylation that is achieved in here. Also, it is obvious that OGT will interact with SOX2 but this interaction itself may impact the interaction of SOX2 with PARP1. A good control in here would be to incubated SOX2, PARP1 and OGT in presence or not of UDP-GLCNAc.*

Forgive our lack of clarity for the affinity purification experiments. Our discovery experiments for the SOX2-protein interactions were performed using well-established, quantitative mass spectrometric methods (Kelihauer et al. MCP, 2015, Sharma et al. Cell Rep, 2014, Schilling et al. MCP, 2012) and is described in the Methods section. Briefly, we used the ratio of a particular protein’s intensity to its respective SOX2 intensity, excluding any TAD peptide signal. We then took the ratio of (ProteinX/SOX2-WT)/(ProteinX/SOX2-S248A). The ratios for all proteins were used to determine the variance of the data, and ultimately which proteins were more enriched by WT or mutant SOX2. The discovery work was carried out on biological replicates and IP-Western analyses used additional, separate biological replicates. We have altered the text in a hope to make these points more clear.

Thank you for the kind words regarding the recombinant work. Because concerns were raised about the possibility of a SOX2:OGT interaction impacting the SOX2:PARP1 interaction, we have substantially redesigned the experiment to remove OGT after the *O-*GlcNAcylation reaction. Briefly, we employed Bio-tagged SOX2 and His-tagged OGT for the *O-*GlcNAcylation reaction, carrying out the reaction +/- the sugar donor UDP-GlcNAc. After the reaction, SOX2 was bound to streptavidin beads and isolated away from the OGT (and UDP in the +UDP reaction). The bead bound SOX2 (+/- *O-*GlcNAcylation, confirmed by Western blot and mass spec) was incubated with GST-tagged PARP1, to assay for interaction. In the new figure, we provide appropriate input controls as well as blots showing that OGT is not present in the SOX2-PARP1 pull down.

*11) There are a number of typos that are apparent in the figures as well as throughout the text (e.g., Figure 1: O-GlcNAc – the "O" is a box; Figure 1, there is no top number on the y-axis, Figure 1 text: μM is μm rather than the "micro", F1a the double O-GlcNAc peptide is mislabeled. Figure 4—figure supplement 1, pictures are shifted… It also appears that tables are missing?*

We thank the Reviewer for pointing out these oversights. We have made the appropriate corrections.

*12) Corresponding mass-spectra of the data presented in Figure 2, Figure 2, Figure 3, Figure 3 are missing and should be provided.*

Again, we thank the Reviewer for pointing out this oversight. We have included interactive spectra from one representative experiment for all peptide PTM forms for all subsequent analyses.

*13) Figure 1, Figure 2, Figure 3: the x-axis is labeled with RT, but there is no indication of the values associated with the retention time. This does not give the reader a good indication of how narrow a window the authors are examining. Moreover, Figure 1 appear to be stretched. Why is there a y ion annotated in Figure 1—figure supplement 4 and Figure 1—figure supplement 5? Is this from a secondary reaction and if so, why are they annotated? Other contaminating peaks that are much larger remain un-annotated.*

Thank you for pointing out the loss of RT labels. The figures have been corrected. We have also added annotation to the mass spectra to identify neutral loss and contamination peaks. The y-ion is present because we use supplemental activation during the ETD reaction. Supplemental activation adds a low level of vibrational energy to dissociate ETD product ions that remain associated due to intramolecular interactions (Swaney et al., Anal Chem, 2007, Good et al., MCP, 2007). While the supplemental activation is low enough to not be the major pathway of unimolecular dissociation, some minor collisional activation dissociation can occur.

*14) O-GlcNAc modification of OCT4 has also been shown to be required for maintaining pluripotency. [Cell Stem Cell 17 May 2012 (doi: 10.1016/j.stem.2012.03.001)] While the authors reference this paper, they do not do so in the context of their OCT4 discussion and this is an important addition for the reader to best understand the state of the field.*

We did not mention the Jang et al. paper in the context of our OCT4 discussion because we have not been able to reproduce their data. We provide our analysis of OCT4 modification as a supplemental figure, so that others interested in the connection between *O*-GlcNAcylation and pluripotency are aware of the contradictory data.

*15) The paper is solid but needs to be rewritten in parts to reflect what was actually tested and verified. Certainly SOX2 plays a key role in the maintenance of pluripotency and O-GlcNAc influences its function. A suggestion for the title may be: O-GlcNAc influences the protein-protein interactions mediated by the pluripotency factor SOX2. It is important that the authors do not leave the impression that this is a unique or sole means by which O-GlcNAc influences this important biological process. Moreover, it is important to be concise – for instance what does 'more pluripotent' stand for? Thus, the manuscript and the figures should be edited carefully throughout.*

These points are well taken. We have gone over the manuscript and made attempts to be more accurate and concise – ‘more pluripotent’ is removed. We struggled with how best to word the title so as not to unintentionally convey the idea that *O*-GlcNAc regulates pluripotency entirely through SOX2. We have altered the title to “SOX2 *O*-GlcNAcylation alters protein-protein interactions and genomic occupancy to modulate gene expression in pluripotent cells”. This title describes the main results without implying that SOX2 is the only relevant OGT target in pluripotent cells. We feel that is necessary to include “SOX2 *O-*GlcNAcylation….” in the title – otherwise it could lead to the incorrect impression that the *O-*GlcNAc modification is not necessarily on the SOX2 itself.

*16) As the authors say, the phenotype of the S248A overexpression mutant is counterintuitive given the requirement for OGT in ESCs. I do not think that different contexts can explain this discrepancy, as the authors suggest, since both studies examine ESC self-renewal. It is more likely that the OGT KO phenotype is due to other O-GlcNac-modified proteins that have opposing effects to SOX2.*

This point is absolutely correct. We have removed this discussion point.